# Groundwater head responses to droughts across Germany

Pia Ebeling[1], Andreas Musolff[1], Rohini Kumar[2], Andreas Hartmann[3], Jan H. Fleckenstein[1,4]

[1]Department of Hydrogeology, Helmholtz Centre for Environmental Research - UFZ, Leipzig, Germany.
[2]Department of Computational Hydrosystems, Helmholtz Centre for Environmental Research - UFZ, Leipzig, Germany.
[3]Institute of Groundwater Management, Technical University Dresden, Dresden, Germany
[4]Bayreuth Center of Ecology and Environmental Research (BayCEER), University of Bayreuth, Bayreuth, Germany.

*Correspondence to*: Pia Ebeling (pia.ebeling@ufz.de)

## Abstract

Groundwater is a crucial resource for society and the environment, e.g. for drinking water supply and dry-weather stream
flows. The recent severe drought in Europe (2018-2020) has demonstrated that these services could be jeopardized by ongoing global warming and the associated increase in the frequency and duration of hydroclimatic extremes such as droughts. To assess the effects of meteorological variability on groundwater heads throughout Germany, we systematically analyzed the response of groundwater heads at 6,626 wells over a period of 30 years. We characterized and clustered groundwater head responses, quantified response time scales, and linked the identified patterns to spatial controls such as land cover and
topography using machine learning. We identified eight distinct clusters of groundwater responses with emerging regional patterns. Meteorological variations explained about 50% of the groundwater head variations, with response time scales ranging from a few months to several years between clusters. The differences in groundwater head responses between the regions could be attributed to regional meteorological variations, while the differences within the regions depended on local landscape controls. Here, the depth to groundwater best explained the time scale of the observed head response, with shorter response
times in shallower groundwater. Two of the clusters showed consistent long-term trends that were not explained by meteorological controls and could be attributed to anthropogenic impacts. Our study contributes to a better understanding of the regional controls of groundwater head dynamics and to the classification of groundwater vulnerability to hydroclimatic extremes.

## 1. Introduction

Groundwater is the largest available freshwater resource worldwide serving numerous water demands such as for drinking, irrigation and industrial water as well as for groundwater-dependent ecosystems, minimum discharges in streams, and dilution of pollutants (Taylor et al., 2013). Droughts can threaten the availability and usability of groundwater to meet these demands and cause severe socioeconomic and ecological impacts (Stahl et al., 2016). The recent multi-year drought in Europe (2018-

2020) has set a new benchmark with extreme socioeconomic damage, resulting in increased public and stakeholder awareness
of the vulnerability of water resources to droughts (Rakovec et al., 2022; Blauhut et al., 2022; Hari et al., 2020). With ongoing climate warming, climatic extremes are intensifying (IPCC, 2023). This includes both increasing frequency, intensity and duration of droughts (Rakovec et al., 2022; Hari et al., 2020; Rodell and Li, 2023) as well as frequency and intensity of extreme precipitation events (IPCC, 2023). This raises the need to develop a thorough understanding of the effects of hydroclimatic variability (including droughts) on groundwater resources and their vulnerabilities to enable an improved knowledge-based
water management.

Droughts are periods with persistent below normal water availability often differentiated by the affected compartments, which the drought signal may propagate through, i.e. meteorological, agricultural (soils), and hydrological droughts (groundwater and surface water) (Van Loon, 2015; Entekhabi, 2023). Generally, when drought signals propagate from the meteorological driving force to groundwater, the landscape acts as a low-pass filter so that the drought response gets attenuated, elongated
and delayed from a more erratic forcing variable to a dynamic with higher memory (Van Loon, 2015; Bloomfield and Marchant, 2013; Kumar et al., 2016). However, this propagation is highly variable across the landscape, with the result that groundwater heads can respond very differently to the driving meteorological forces (Bloomfield and Marchant, 2013; Kumar et al., 2016). Previous studies have investigated the controls of groundwater dynamics at different spatial scales (local to integral catchment scales) and temporal representations (daily heads to monthly anomalies). They have highlighted the
importance of hydrogeological conditions and the well location, more specifically, the aquifer type (Bloomfield et al., 2015; Hellwig and Stahl, 2018), confinement status (Haaf et al., 2020; Bloomfield and Marchant, 2013), the hydrological conductivity (Hellwig et al., 2020), the unsaturated zone thickness or depth to groundwater (Bloomfield et al., 2015; Haaf et al., 2020; Lischeid et al., 2021; Wossenyeleh et al., 2020; Kumar et al., 2016), the distance to stream (Haaf et al., 2020), and the location along the topographic gradient (Haaf et al., 2020; Schuler et al., 2022; Rinderer et al., 2017). Haaf et al. (2023)
and Peters et al. (2006) also highlighted the non-linearity of processes, which can cause different controls of groundwater dynamics to dominate during wet and dry conditions or groundwater recharge and discharge. Nevertheless, uncertainties in future groundwater resource availability (Marx et al., 2021; Wunsch et al., 2022; Kumar et al., 2025; Reinecke et al., 2021; Berghuijs et al., 2024) are not only related to uncertainties in climate projections (e.g., Naumann et al., 2021) and model implementation (e.g., Kumar et al., 2025; Reinecke et al., 2021), but also to the challenge to fully understand spatial variability
of groundwater head responses across locations (e.g., Lischeid et al., 2021). Consequently, we argue that it is still insufficiently known how the different meteorological and landscape controls play out together to create spatial and temporal variability in groundwater heads and to what extent the controls can be generalized.

Large-sample data-driven analyses of groundwater responses to climatic drivers and underlying controls of spatial variability can be a promising way to further elucidate this interplay in controls. Standardized indicators create comparability across
stations, regions and compartments of the hydrological cycle. Often meteorological drought indicators are used to assess

hydrological droughts as the data is comprehensive and easily accessible (Van Loon, 2015; Bachmair et al., 2016), although they are not directly transferable into groundwater droughts observed locally at groundwater wells (Kumar et al., 2016). In contrast, large-sample analyses of groundwater droughts are challenged by the limited availability of consistent groundwater head data sets, as the indicators are sensitive to the covered time periods (Van Loon, 2015; Bloomfield and Marchant, 2013;

Bachmair et al., 2016). Groundwater data sets often have systematic gaps, cover different periods and sampling frequencies, and/or are not fully accessible (Bikše et al., 2023; Barthel et al., 2021). This limited data availability and consistency often hampers large-scale and comparative groundwater analysis (Barthel et al., 2021; Haaf et al., 2020), the understanding of the spatial variability in groundwater responses and drought propagation, and the inference of drought vulnerability.

The vulnerability of a system can be interpreted as its disability to maintain or return to its state in the face of particular stresses.

For groundwater heads, the most obvious example of stress is a meteorological anomaly, such as an extreme meteorological drought. However, hydroclimatic extremes can have different manifestations, e.g. short duration with a high intensity or long duration with a lower intensity (Hari et al., 2020; Hosseinzadehtalaei et al., 2020; Westra et al., 2014; e.g., Christian et al., 2023). Moreover, as indicated above, groundwater responses and associated response time scales are highly variable in space (e.g., Lischeid et al., 2021). The different manifestations of groundwater head responses suggest different vulnerabilities of

their corresponding groundwater systems with implications, for example, for surface-groundwater interactions, ecosystems or groundwater management, and with distinct sensitivities regarding expected changes in climate. Therefore, a better understanding of types of vulnerability and their controls is required.

In this study, we perform a large-sample data-driven analysis of groundwater head responses to meteorological anomalies to understand their spatial variability and controlling factors. We use a consistent large-sample dataset of 6,626 monthly

groundwater head time series of 30 years across Germany to identify similarities and differences in groundwater responses and quantify time scales of propagation from meteorological anomalies to groundwater. Finally, we link the response patterns to spatial controls including climatic and landscape properties. On this basis, we can classify different vulnerabilities of groundwater to meteorological droughts and discuss implications for water management and ecology.

## 2. Methods

### 2.1. Data

The groundwater head data used in this study are monthly mean groundwater head time series across Germany provided by journalists of the CORRECTIV.Lokal network for the period from 1990 to 2021 (Donheiser, 2022; Joeres et al., 2022). CORRECTIV is a non-profit network of journalists that collected the groundwater head time series from the different environmental Federal state authorities responsible for groundwater monitoring in order to report about the groundwater

conditions during the recent drought years (Joeres et al., 2022). They homogenized the data by aggregating the original observations (heterogeneous, partly daily resolution) to monthly resolution and provide those in a free repository (for details refer to Donheiser, 2022). This implies that we have a consistent monthly time scale for the analysis, at the cost of having less control on the preprocessing of the original data. For the initial selection of stations of our study, we used the 6,677 stations identified by CORRECTIV based on the criteria: having data for at least 95% of the months, showing no shifts in the head

time series, and having station coordinates (Donheiser, 2022).

We filled gaps in monthly heads with linear interpolation without extrapolation using the function *na.interp* (R package forecast, Version 8.21; Hyndman and Khandakar, 2008; Hyndman et al., 2023). For 4,197 of the wells, at least one missing value had been filled, out of which 69.1% of the wells (2,899) had a maximum gap length filled less than 3 months and overall the maximum gap length was 19. We finally selected 6,626 stations covering the complete period from January 1991 to

December 2020.

The wells of the data set (Fig. 1) tend to be located in highly productive porous aquifers (60.3% of wells; aquifer types from IHME1500; BGR and (eds.), 2014), coarse sediments (i.e. gravels and sands, lithology from IHME1500; BGR and (eds.), 2014) and medium to high hydraulic conductivities (62.5%; BGR and SGD, 2016). Moreover, wells are predominantly in shallow aquifers, i.e. 50.7% of wells have a mean groundwater depth <5m and 74.5% <10m (based on *mean_gwdepth*, see

Table 2). Such a sampling bias is typical for groundwater wells, as these locations are more relevant for water management (e.g., Barthel et al., 2021). The majority of wells (50.2%) are located in agricultural areas, 27.0% in urban and 21.7% in forested areas (EEA, 2019b). About one-third of the wells are located in areas classified as riparian zones (EEA, 2021). Regionally, particularly high densities of wells are found in the city of Berlin (388 wells, i.e. 0.44 wells km$^{-2}$) and in the southwestern Upper Rhine Plain (Ger.: Oberrheinische Tiefebene), whereas low densities (<0.01 wells km$^{-2}$) are found in the federal states

of Mecklenburg-Vorpommern, North Rhine-Westphalia and Bavaria (Fig. 1). No data is available for the federal states of Saarland, Bremen and Hamburg.

For time series of meteorological drivers at each well location, we extracted daily time series of climate variables (i.e., precipitation, maximum, minimum and average air temperature) from the gridded (approx. 1km resolution) products derived based on measurements from the German weather service (DWD; Boeing et al., 2022; Zink et al., 2017) from 1971 until 2020.

Potential evapotranspiration was calculated by the approach from Hargreaves and Samani (1985). Daily time series were then aggregated to monthly mean values.

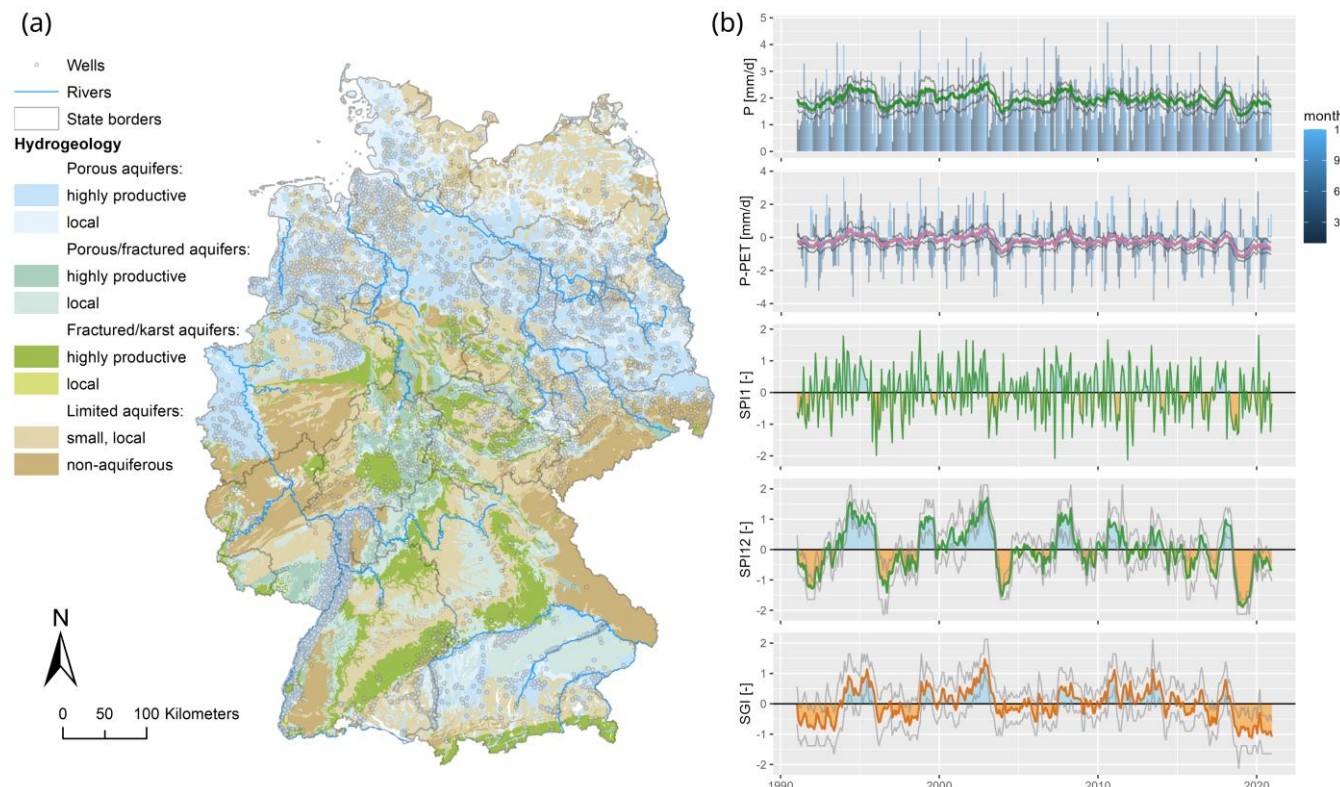

**Figure 1: Study area with groundwater wells (Donheiser, 2022), hydrogeological classes of aquifer types (HY1000 v1 © BGR 2019; BGR (2019)), and major rivers (from Strahler order 6; EEA, 2020) (a) and time series of monthly precipitation and accumulated precipitation of preceding 12 months (P$_{12}$, green line), monthly P-PET and 12 months accumulated P$_{12}$-PET$_{12}$ (pink line), SPI$_1$, SPI$_{12}$ and SGI as spatial average across wells (b). The thin gray lines in panel b indicate the 25th and 75th percentile across wells. P - precipitation; PET - potential evapotranspiration; SGI - Standardized Groundwater Index, SPI - Standardized Precipitation Index**

## 2.2. Characterizing anomalies

To characterize the groundwater responses with a focus on droughts and to ensure comparability across locations (Van Loon, 2015), we standardize the groundwater and meteorological time series representing anomalies. Anomalies generally describe deviations from average conditions, with positive values indicating relatively wetter and negative values indicating drier conditions.

### 2.2.1. Groundwater

Groundwater head anomalies were characterized based on median groundwater heads of each month using the non-parametric Standardized Groundwater Index (SGI; Bloomfield and Marchant, 2013). This approach assesses anomalies in a groundwater head time series by comparing the value for a given month to the distribution of all values for the same month, effectively eliminating seasonal variability in groundwater heads. More specifically, we used normal scores transform by assigning

equally spaced probabilities to the ranked groundwater heads of each month of a given time series separately and applying the inverse normal cumulative distribution function to get standard normal distributed values (mean of zero, standard deviation of one, Bloomfield and Marchant, 2013). This implies that probabilities range from $(1/2n)$ to $(1-1/2n)$ with $n=30$ due to the 30 values for each month and the corresponding SGI values from the normal distribution are sorted according to the ranks of the groundwater heads, i.e. the lowest SGI is assigned to the lowest groundwater head of the respective month. This non-parametric standardization is particularly suitable for irregular and different distributions, typical for groundwater heads, as it does not require fitting different distribution functions that hamper comparability of resulting SGI time series (Bloomfield and Marchant, 2013).

To characterize groundwater droughts, we calculated different intrinsic properties of the SGI time series (Table 2). Firstly, we determined the autocorrelation length, which we defined as the maximum lag, where both the lag itself and all smaller lags exhibit correlation coefficients greater than 0.11 referring to approx. 5% significance level (Bloomfield and Marchant, 2013). Secondly, we identified groundwater drought events defined as consecutive months with SGI < -1 (i.e. a probability of <15.9% according to the standard normal distribution). We then calculated the number, average duration and severity of drought events for fully covered events within the 30-year time period. The event severity is defined as the integral event anomaly determined by cumulative SGI values during the event. Thirdly, we quantified monotonic trends in the SGI time series applying Mann-Kendall trend and Sen's slope analysis. We used functions *mk.test* (p-value<0.01) and *sens.slope* from R package *trend* (version 1.1.5; Pohlert, 2023).

### 2.2.2. Meteorology

Meteorological anomalies generally represent deviations from average conditions at a specified location and time, e.g. precipitation deficits or surplus. To characterize them, we computed the Standardized Precipitation Index (SPI, McKee et al., 1993) from monthly mean precipitation and the Standardized Precipitation Evapotranspiration Index (SPEI, Vicente-Serrano et al., 2010) from monthly mean differences between precipitation and potential evapotranspiration. The SPI and SPEI were estimated based on the monthly mean values, using the same non-parametric standardization as for the SGI comparing values to all other values of the same month (details described in Sect. 2.2.1).

To represent meteorological anomalies across longer antecedent time periods and thereby account for the different relationships that SGI and meteorological variables may have (Kumar et al., 2016; e.g., Bloomfield and Marchant, 2013), we calculated the $SPI_{acc}$ and $SPEI_{acc}$ for different accumulation periods (acc) of precipitation and precipitation-potential evapotranspiration preceding the corresponding month by up to 132 months (11 years). For example, the $SPI_3$ is calculated based on precipitation sums of three months and thus characterizes the precipitation anomaly of the past three months.

The advantage of the SPEI over the SPI is that it is sensitive to temperature effects on drought severity and thus global warming (Vicente-Serrano et al., 2010; Van Loon, 2015). However, we acknowledge that the period of 30 years covered in this study is short to robustly represent climate change effects as generally trend analysis of groundwater heads and drought indicators have been shown to be sensitive to the covered periods (Bloomfield and Marchant, 2013; Hellwig and Stahl, 2018; Lischeid et al., 2021).

### 2.3. Clustering of groundwater anomalies

To find regional similarities and differences in the responses of the groundwater wells, we clustered the SGI time series using k-means as an unsupervised machine learning algorithm. We applied the *kmeans* function implemented in R's stats package (R Core Team, 2023) using Euclidean distance to quantify (dis-)similarities. The Euclidean distance measures (dis-)similarity based on the squared differences of two SGI time series, making it sensitive to extreme differences and temporal shifts but also computationally efficient.

We selected the optimal number of clusters (k) based on the average silhouette distance, which measures the compactness of the clusters and the separation from other clusters based on dissimilarity of the members within one cluster and to the members of the nearest neighboring cluster. The silhouette coefficient ranges from [-1, 1] with one being the optimal value, zero indicates that the member is placed exactly in between two clusters, while a negative value indicates the identity would rather belong to a neighboring cluster. To compute the distances, we used the *silhouette* function from R package *cluster* (version 2.1.4; Maechler et al., 2022) with 25 iterations for starting points of cluster centers (nstart=25). Across different k up to 20, the silhouette distance showed local maxima at k=2, 5 and 8 with average distances around 0.11 (in decreasing order, Fig. S1).

To take a confident decision, we additionally consulted the total within-cluster sum of squares as a measure of cluster compactness in a scree plot. This method is known as "elbow method", where the inflection point indicates the optimal number of clusters. The results from the analysis of silhouette distance and "elbow method" are shown in the supporting material (Fig. S1-S3). Finally, we decided for eight clusters as an optimum between differentiating and generalizing the individual identities.

### 2.4. Response times of groundwater to meteorological drivers

To investigate propagation of meteorological drought to groundwater drought, we calculated cross-correlation between SGI time series and meteorological drivers (SPI, SPEI) in positive direction (i.e. meteorological forcing preceding the groundwater response). The cross-correlation is calculated for the different accumulation periods up to lag times of 5 years (60 months) using *ccf* function in R. The maximum cross-correlation coefficient (cc) result yielded the optimal accumulation time (acc) and corresponding lag time (lag; Table 2).

To quantify trends in meteorological drivers in comparison to the groundwater SGI, we calculated Mann-Kendall trend and Sen's slope on standardized meteorological variables (SPI, SPEI) and on the residuals from a linear regression between SGI and $SPI_{acc}$ (and $SPEI_{acc}$) applying the cross-correlation results of each well. Assuming a simple linear relationship between SGI and $SPI_{acc}$ and $SPEI_{acc}$ respectively, this provides an estimate of trends not reflected in the meteorological driving forces

which could hint towards other relevant drivers, such as anthropogenic impacts.

**Table 1: Groundwater response characteristics, including intrinsic properties and linkages between meteorological drivers and groundwater responses. For each parameter, the corresponding method used to calculate it, the unit**
**and the minimum, median and maximum values across all groundwater wells are provided (although the latter constitute results). Note: the minimum (Min), median, maximum (Max) values in brackets refer to the characteristics regarding the SPEI, instead of the SPI. Number of wells n=6,626.**

| Category | Parameter | Method | Unit | Min | Median | Max |
|---|---|---|---|---|---|---|
| Intrinsic | acf_lag | Autocorrelation length | year | 0.08 | 1.75 | 11.42 |
| | event_n | Number of events fully covered within the 30-year time series, for details see text | | 0 | 13 | 44 |
| | event_length | Average event duration | month | 0.08 | 0.5 | 14 |
| | event_cumm | Average drought severity defined as cumulative SGI values (integral of the anomaly time series) during an event | | -88.61 | -5.12 | -1.04 |
| | sgi.sen_slope | Trend in SGI time series as Sen's slope, for details see text | $month^{-1}$ | -8.9 E-3 | -8.8 E-4 | 8.9 E-3 |
| Meteo-groundwater linkages | acc | accumulation period in SPI (SPEI) with highest cross-correlation result (cc) to the SGI times series | month | 1 (1) | 13 (13) | 132 (132) |
| | cc | maximum correlation coefficient between SGI and SPI (SPEI) corresponding to acc and lag time | | 0.041 (-0.075) | 0.70 (0.71) | 0.93 (0.95) |
| | lag | lag time between SGI and $SPI_{acc}$ ($SPEI_{acc}$) to reach maximum cross-correlation | month | 0 (0) | 0 (0) | 60 (60) |
| | respt | response time of groundwater anomalies to meteorological anomalies (SPI and SPEI), defined as the center of meteorological anomalies, i.e.respt = 0.5 * acc + lag | month | 0.5 (0.5) | 7 (6.5) | 126 (126) |
| | resid_sen | Trend in SGI-$SPI_{acc}$ (SGI-$SPEI_{acc}$) residuals as Sen's slope on residuals from linear regression between SGI and $SPI_{acc}$ time series (considering *acc* and *lag*) | $month^{-1}$ | -8.7 E-3 (-7.2 E-3) | -1.9 E-4 (9.0 E-4) | 1.1 E-2 (1.0 E-2) |

## 2.5. Spatial controls of groundwater responses

### 2.5.1. Spatial properties

To investigate controls of groundwater drought response patterns, we determined several spatial properties including topographical, climatic, land cover and hydrogeological characteristics as well as the relative location of the well in the landscape. The total set includes 26 parameters. Details on the calculated properties is provided in Table 2. To quantify collinearity among the properties, we calculated pair-wise Spearman rank correlations (Fig. S4).

**Table 2: Spatial properties calculated for each groundwater well. For each parameter, the corresponding method**
**used to calculate it (including the data source if applicable), the unit and the minimum, median and maximum values within the data set are provided.**

| Category | Parameter | Method | Unit | Min | Median | Max |
|---|---|---|---|---|---|---|
| Topography | dem | elevation extracted from EU DEM with 100m resolution (Ebeling et al., 2022; EEA, 2013) | m | -4.5 | 95.1 | 920.3 |
| | slo | topographic slope based on DEM (Ebeling et al., 2022; EEA, 2013) | ° | 0 | 0.8 | 21.9 |
| | twi | topographic wetness index based on DEM (Ebeling et al., 2022; EEA, 2013; Beven and Kirkby, 1979) | | 10.5 | 14.2 | 28.2 |
| Climate | P_mm | Mean annual precipitation (1971-2020) | mm | 466 | 698 | 2062 |
| | PET_mm | Mean annual potential evapotranspiration (1971-2020) | mm | 543 | 773 | 882 |
| | AI | Mean annual aridity index AI = PET_mm / P_mm | | 0.34 | 1.19 | 1.77 |
| | P_SI | Precipitation seasonality index as the sum of absolute differences between the mean monthly and one twelfth of the annual precipitation (P_mm/12) normalized by P_mm | | 0.07 | 0.17 | 0.39 |
| | PET_SI | Potential evapotranspiration seasonality index as the sum of absolute differences between the mean monthly and one twelfth of the annual PET (PET_mm/12) normalized by PET_mm | | 0.60 | 0.67 | 0.72 |
| Relative location | mean_gwdepth | difference between elevation (dem) and the mean groundwater level based on the filled head time series analyzed in this study | m | -11.3 | 4.9 | 141.6 |
| | river_dist_m | Horizontal distance to the closest stream from EU-Hydro (EEA, 2020) | m | 0.14 | 518.16 | 10952.35 |
| | lake_dist_m | Horizontal distance to the closest lake from EU-Hydro (EEA, 2020) | m | 0 | 1816 | 20324 |
| | dsd$_{order}$ | distance to stream and catchment divide from the data set Multiorder Hydrological Position (MOHP, Nölscher et al., 2022a, | m | 930 | 48526 | 133638 |

| Category | Parameter | Method | Unit | Min | Median | Max |
|----------|-----------|--------|------|-----|--------|-----|
| | lp$_{order}$ | b). "Order" refers to the stream order considered for assessing the point location relative to the stream network. To reduce redundancies only orders 2, 4, and 6 were used, note: values from order 6 lateral position from the MOHP (Nölscher et al., 2022a, b), see also dsd$_{order}$. The lateral position indicates proximity to stream relative to the watershed divide, with 0 - at stream, 1 - at boundary | | 0.0003 | 0.4 | 1 |
| | sd$_{order}$ | stream distance from the MOHP (Nölscher et al., 2022a, b), see also dsd$_{order}$ | m | 30 | 14072 | 91454 |
| Hydrogeo-logy | kf_rank | Hydrologic conductivity rank order based on L_KF class from HÜK200 (BGR and SGD, 2016) for upper aquifer at well location. Ranks sort the hydraulic conductivities from high (L_KF class 2) to extremely low (L_KF class 7). Mixed classes were assigned mean numeric values of the corresponding classes, e.g. L_KF class 9 represents a mix of class 3 and 4 and thus got the value 3.5 | | 2 | 3 | 7 |
| Land cover | y18_artificial_10km | fraction of artificial within a 10km buffer around the well (CODE 1 from level 1 classes) from CORINE Land cover map 2018 (EEA, 2019b) | | 0.000 | 0.098 | 0.977 |
| | y18_agriculture_10km | fraction of agriculture (CODE 2), see y18_artificial_10km (EEA, 2019b) | | 0.000 | 0.5585 | 0.957 |
| | y18_forest_10km | fraction of forest (CODE 3), see y18_artificial_10km (EEA, 2019b) | | 0.000 | 0.237 | 0.945 |
| | y90_mining_frac_10km | fraction of mining within a 10km buffer around the well (Code 13 from level 2 classes) from CORINE Land cover map 1990 (EEA, 2019a) within a 10km buffer around the well | | 0 | 0 | 0.215 |
| | hy_3km_intersect | Intersection with mining areas indicated in HY1000 map (BGR, 2019) and 3km buffer around | boolean | 0 | 0 | 1 |

## 2.5.2. Machine learning

To identify spatial controls on the observed groundwater responses, we first trained different random forest (RF) classification and regression models (Breiman, 2001) to predict the identified clusters (Sect. 2.3) and groundwater response times (Sect. 2.4)

of the 6,626 wells from the 26 spatial controls (Sect. 2.5.1). Second, we use interpretable machine learning tools to reveal insights into the relationships learned by the machine learning models. More specifically, we apply the global model-agnostic methods permutation feature importance and partial dependence plots (PDP), allowing us to investigate average model

behavior and thus discuss prevalent relationships. RFs are particularly well-suited for efficiently handling large datasets, managing collinearity among descriptors through random feature selection, and identifying complex non-linear relationships without a priori assumptions. Moreover, they are robust to outliers and noise due to their ensemble approach averaging across trees (Breiman, 2001).

In detail, to predict the clusters of groundwater responses and groundwater response times, we train (1) two RF classification models for (i) all clusters, (ii) clusters with regional prevalence, and (2) RF regression models for the characteristics acf_lag, acc, respt, and resid_sen for SPI and SPEI each (Table 1). We used 5-fold cross-validation to evaluate the models, i.e. five iterations for each model. Model performance was evaluated on the five sets of test data using the mean accuracy (percentage of correct classifications) for classification and the mean coefficient of determination $R^2$ for regression models.

Feature importance was evaluated by the relative increase in model error with permutation using the classification error (percentage of incorrect classifications) and root mean square error for regression as loss functions. Each feature was permuted ten times for each of the five test data sets of the cross-validation to acquire robust results. Subsequently, the importance results were aggregated across the five resamplings providing an average and range of importance for each feature. For selected features and models, we created partial dependence plots (PDP) to analyze the effect of features on the model output by using the RF models trained on the full dataset. For model training and evaluation we used the *mlr3* package in R (version 0.17.0, Lang et al. 2019) and for model agnostics we used the *iml* package (version 0.11.1, Molnar 2018).

## 2.6. Classifying vulnerability to droughts

Vulnerability of groundwater systems to droughts is here defined based on the response times to meteorological anomalies, with different possible implications for management and ecology. We understand response times as the time delay in the propagation of the anomaly signal from the driving force to the responding variable. The accumulation time includes this temporal delay as it accounts for meteorological anomalies preceding the corresponding time (Sect. 2.4). The center of these anomalies can be considered as half of the accumulation time. We thus calculated the response times from the SPEI (respt$_{SPEI}$) by taking half of the optimal accumulation time (acc$_{SPEI}$) and adding the corresponding identified cross-correlation time lag (lag$_{SPEI}$) (Table 1). The respt$_{SPEI}$ can thus be interpreted as a response time from center to center (or in other words, peak to peak). Finally, we classified the wells into fast, medium and slow responding groundwater systems based on three quantiles of the distribution of respt$_{SPEI}$ (i.e. the 33$^{rd}$ and the 67$^{th}$ percentiles). These classes can serve as an important element in vulnerability assessments of groundwater to meteorological droughts with different characteristics.

## 3. Results

### 3.1. Variability in groundwater responses

Overall, groundwater head responses were diverse both in terms of temporal and across-site variability. Regional patterns emerged with distinct groundwater drought responses grouped into eight clusters based on the similarities in the SGI time series (Fig. 2). Two clusters spread across the entire country (clusters lt_inc and lt_dec), while two clusters each predominated in three distinct regions, i.e. in northeastern (ne_lf, ne_hf), northwestern (nw_hf, nw_lf) and southern (sw_lf, sw_hf) Germany, respectively. More closely, the cluster lt_inc, although scattered across Germany, was still more prevalent around Berlin and in the Upper Rhine Plain. The number of wells within each cluster ranged from 570 (8.6%, cluster lt_dec) to 1179 (17.8%, cluster sw_hf). We named the eight clusters according to their dominant characteristics, referring to the clusters' regional prevalence (nw - northwest, ne - northeast, sw - southwest), their intrinsic frequency in change of the SGI time series (lf - low frequency, hf - high frequency) or the dominant long-term trend in the SGI time series (lt_inc - increasing trend, lt_dec - decreasing trend). The characteristic response patterns are described in the following paragraphs and summarized in Table 3.

Across all wells, about 36% of the wells reached their driest conditions, based on the minimum mean annual SGI, in the last two years of the covered 30-year period (i.e. 2019 and 2020), and an additional 4.2% in the year 2018. Across the 30 years, we found 48.2% of the wells to have significant negative trends in the monthly SGI, indicating drying processes, and 26.2% positive trends, indicating increasing wetness. Across clusters, the number of positive and negative trends in SGI varied, with cluster lt_inc with long-term increasing heads being dominated by positive (98.4%) and cluster lt_dec with long-term decreasing heads being dominated by negative (100%) trends, respectively. All other clusters were more balanced, with a maximum of 76.8% of cluster wells (in the case of cluster nw_lf with negative trends) in one trend class (Fig. S8, Table 3). In contrast, in precipitation anomalies in the form of the monthly $SPI_1$ only 11.4% of well locations showed negative trends and the majority had no significant trends (88.5%) and <0.1% showing positive trends. The time series of the meteorological anomaly $SPEI_1$ have 60.2% negative trends, 39.8% non-significant, and <0.1% positive trends. For longer accumulation periods of the anomalies, the meteorological trends shift towards more negative trends, e.g. 56.9% for $SPI_{24}$ and 92.5% for $SPEI_{24}$. The SPI and SPEI show systematic differences, with SPEI showing more pronounced negative trends, linking to higher SPEI values during the 1990s and lower values in the last decade of the time series (Fig. S5).

Intrinsic time series properties also varied across sites and clusters (Table 3). Autocorrelation length (acf_lag) across the SGI time series ranged from 1 month to 11.4 years, with a median of 1.75 years (Table 1, Fig. S8). The distribution of mean drought durations across stations was right-skewed with a median of 3.6 months and average drought severity of -5.12, while the median number of drought events during the 30 years was 13. Across clusters, three clusters (nw_hf, ne_hf and sw_hf) had considerably shorter autocorrelation lengths on average (median *acf_lag* between 0.5 and 1.5 years) than the other three clusters (nw_lf, ne_lf, sw_lf) dominating within the same region (*acf_lag* between 1.5 and 2.1 years; see Fig. S8). This means that they

have a higher frequency in their SGI variability (hf) compared to their regional counterparts with a lower frequency (lf). Similarly, the high-frequency clusters with shorter autocorrelation lengths (nw_hf, ne_hf and sw_hf) had more drought events (median of 14–22 events compared to 8–9) with shorter mean drought duration (median *event_length* between 2.7 and 3.5 compared to 5.0 and 5.9 months) and lower mean drought severity (median *event_cumm* above -5 compared to below -7) than the low-frequency clusters nw_lf, ne_lf, sw_lf. The cluster describing the long-term trends (lt_inc, lt_dec) on the other hand had the highest autocorrelation lengths on average (median *acf_lag* of 4.67 and 5.00 years respectively) although generally also covering a broad range of values.

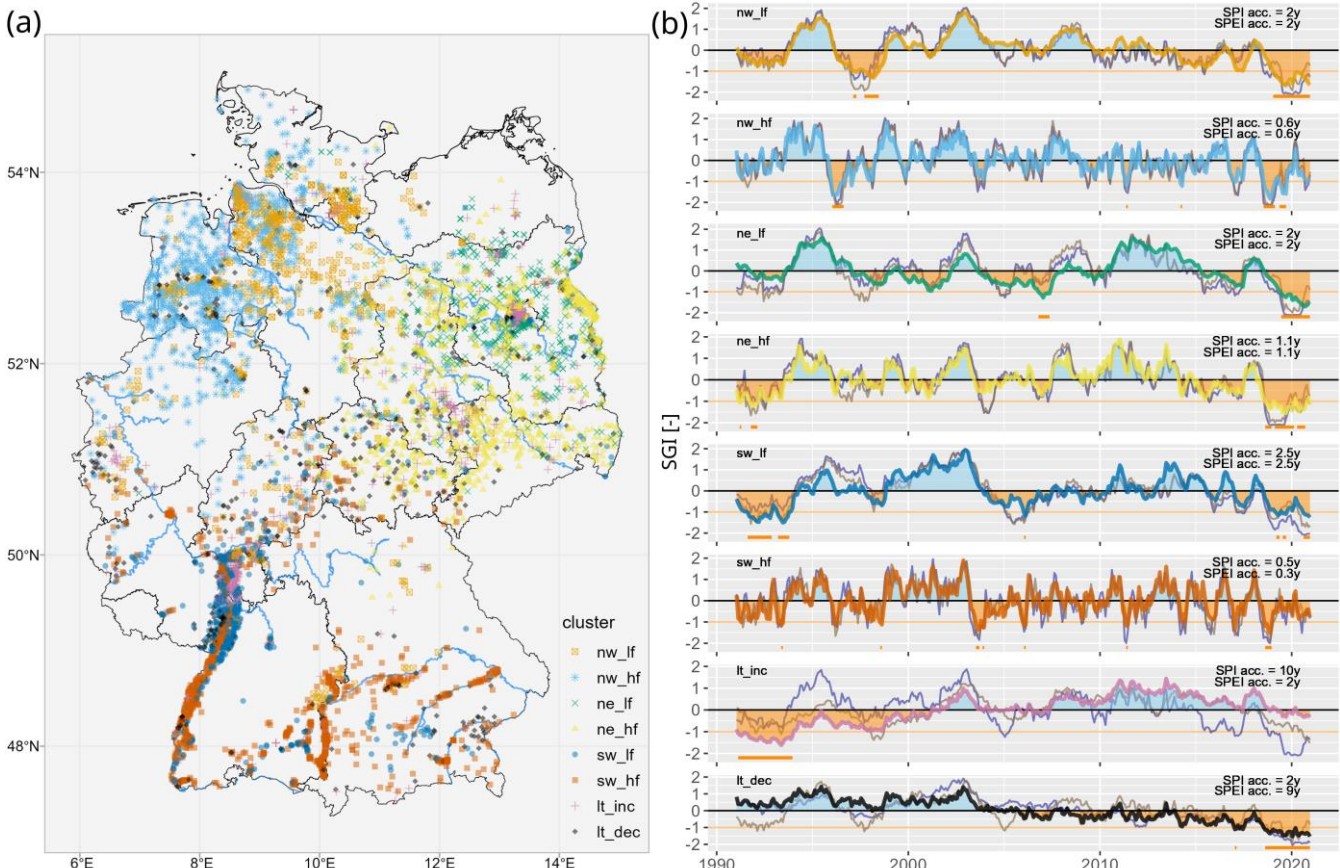

**Figure 2: Spatial distribution of clusters of groundwater head anomalies within Germany and major rivers (blue lines, Strahler order 6 and higher; EEA, 2020) (a) and time series of groundwater head anomaly (SGI), precipitation anomaly (SPI$_{acc}$) and precipitation-evapotranspiration anomaly (SPEI$_{acc}$) of the clusters (i.e. mean across cluster members) (b). Orange shading in (b) refers to negative SGI values (i.e. relatively dry conditions), and blue shading to positive SGI values (relatively wet conditions). The orange line at SGI=-1 indicates the threshold used for drought events, while orange segments below indicate occurrences of drought events for cluster means. The time series in gray and purple denote the SPI$_{acc}$, and SPEI$_{acc}$, resp., of the highest cross-correlation derived for the cluster means. Cluster names according to their regional prevalence (nw - northwest, ne - northeast, sw - southwest) and intrinsic SGI pattern (hf - high frequency, lf - low frequency, lt_dec - long-term decrease, lt_inc - long-term increase; see Table 3).**

### 3.2. Response times of groundwater heads to meteorological drivers

Relationships between individual SGI time series and the meteorological variables varied strongly with cross-correlation coefficients from around 0 up to 0.95 (Fig. 3, Fig. S8), with a median around 0.70 for both SPI and SPEI. This corresponds to 50% of the variance of the groundwater head anomalies being explained by the meteorological drivers on average. The optimal accumulation periods yielding maximum cross-correlation had a median of 13 months. The corresponding time lag between the time series at maximum cross-correlation was mostly zero, nevertheless it is important to note that there is a delay from the driver to the groundwater response implicitly included in the corresponding accumulation period considering the antecedent meteorological variables.

The cross-correlation results indicate that the high-frequency clusters (nw_hf, ne_hf and sw_hf) with shorter autocorrelation lengths (Sect. 3.1) were characterized by shorter optimal accumulation periods with median $acc_{SPEI}$ ranging from four months (sw_hf) to one year (ne_hf) (Fig. 3, Fig. S8, Table 3), i.e. representing systems with shorter system memories. In contrast, for the low-frequency clusters (nw_lf, ne_lf, sw_lf) the median accumulation times were about two years. This is also reflected in cluster means, with accumulation times $acc_{SPI}$ and $acc_{SPEI}$ of the high-frequency clusters being considerably lower (between 0.3 and 1.1 years) than their regional low-frequency counterparts (2-2.5 y, Fig. 2b). The cross-correlations were weakest for the cluster with a long-term increase in SGI (lt_inc), particularly for the precipitation-evaporation index (SPEI) with a median coefficient of 0.41 across cluster members, while the median was 0.60 for the precipitation index (SPI, Table 3, Fig. S7 and S8). Cluster lt_dec with long-term decreasing SGI, in contrast, had higher median cross-correlations for the SPEI (0.70) compared to the SPI (0.61, Fig. S8).

This weaker link of trend clusters lt_inc and lt_dec with the meteorological driver is also reflected in predominant trends in the residuals between the SGI and corresponding meteorological SPEI time series (see Fig. 5 a, Table 3). Cluster lt_inc has 98.4% positive trends in the SGI-SPEI$_{acc}$ residuals and 88.2% in the SGI-SPI$_{acc}$ residuals, whereas cluster lt_dec has 94.6% negative trends in the SGI-SPI$_{acc}$ residuals and 84.7% in the SGI-SPEI$_{acc}$ residuals.

**Table 3: Selected groundwater response characteristics per cluster. The values provided are medians of the cluster. For the SGI trends and the trends in residuals between SGI and SPEI$_{acc}$, the majority of the direction of trend is indicated with "-" for negative, "+" for positive and "ns" for non-significant trend together with the corresponding fraction. The parameters indicate n – number of samples in the cluster, acf_lag – autocorrelation length in years, event_length – average groundwater drought event length in months, accSPI/accSPEI – optimal accumulation length from crosscorrelation, ccSPI/ccSPEI – crosscorrelation coefficient, resptSPI/resptSPEI – response times of**

**groundwater to meteorology. Further details on the parameters are given in Table 1 and in the text, distributions of values within the clusters are visualized in Fig. 3, 5 and S8.**

| Cluster | Region | Characteristic | n | acf_lag | event_length | majority SGI trend (-) | $acc_{SPI}$ ($acc_{SPEI}$) | $cc_{SPI}$ ($cc_{SPEI}$) | $respt_{SPI}$ ($respt_{SPEI}$) | residual trend (-) |
|---------|--------|----------------|---|---------|--------------|------------------------|----------------------------|--------------------------|--------------------------------|--------------------|
| nw_lf | northwest | Low frequency, slow response, long memory | 594 | 1.5 | 5.9 | - (0.77) | 24 (24) | 0.79 (0.79) | 13 (13) | + (0.43) |
| nw_hf | northwest | High frequency, fast response, short memory | 918 | 0.83 | 2.7 | - (0.72) | 6 (6) | 0.72 (0.74) | 3.5 (3.5) | ns (0.45) |
| ne_lf | northeast | Low frequency, slow response, long memory | 584 | 2.1 | 5.5 | - (0.51) | 24 (24) | 0.70 (0.73) | 13 (13) | + (0.42) |
| ne_hf | northeast | High frequency, fast response, short memory | 1099 | 1.5 | 3.5 | - (0.48) | 12 (12) | 0.72 (0.74) | 6.5 (6) | + (0.46) |
| sw_lf | southwest | Low frequency, slow response, long memory | 752 | 2.1 | 5.0 | ns (0.38) | 24 (21) | 0.75 (0.72) | 12.3 (11.5) | + (0.76) |
| sw_hf | southwest | High frequency, fast response, short memory | 1179 | 0.5 | 2.7 | ns (0.43) | 5 (4) | 0.68 (0.69) | 2.5 (2.5) | + (0.54) |
| lt_inc | national | Long-term increase beyond meteorology | 930 | 4.7 | 4.4 | + (0.98) | 54 (30) | 0.60 (0.41) | 29 (18) | + (0.98) |
| lt_dec | national | Long-term decrease beyond meteorology | 570 | 5.0 | 3.7 | - (1.00) | 18 (24) | 0.61 (0.70) | 10.5 (13) | - (0.85) |

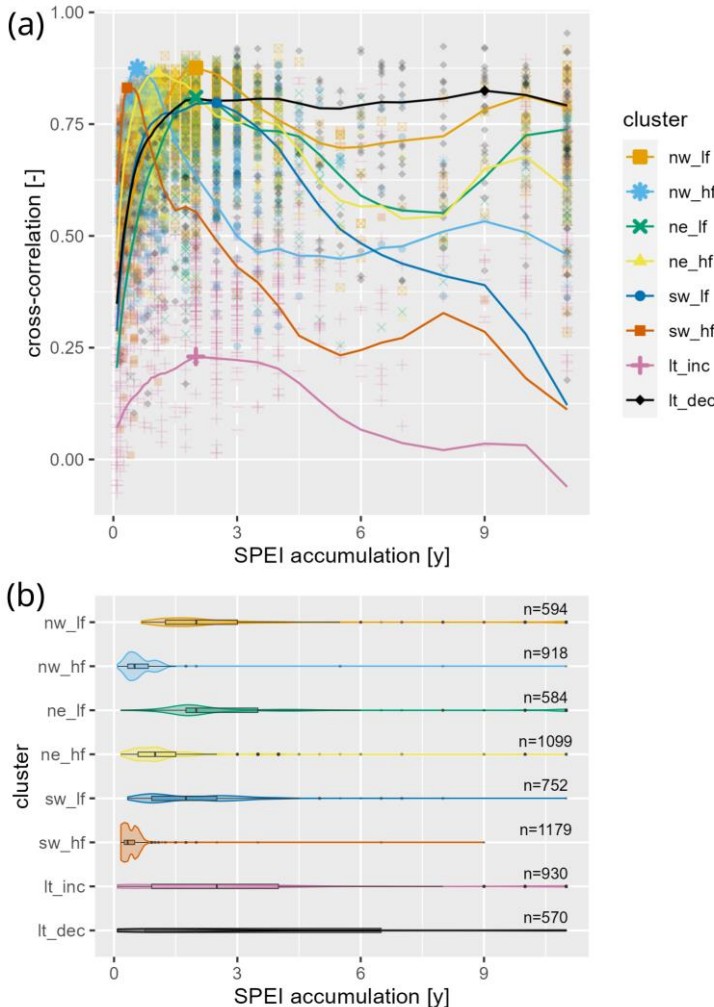

Figure 3: Groundwater memory time scales to meteorological drivers: (a) Relationship between cross-correlation between groundwater anomaly SGI and precipitation-evapotranspiration anomaly SPEI with different accumulation periods for cluster means (lines) and maximum cross-correlations of cluster means and of individual wells (points) and (b) distribution of optimal SPEI accumulation times of wells within the clusters as violin plots with additional boxplots visualizing summary statistics (median, the 25th and 75th percentiles). n - number of wells per cluster.

## 3.3. Spatial controls of groundwater head responses

Random forest (RF) models were trained and evaluated to identify controls of groundwater response patterns and time scales to meteorological drivers. Results of selected models including the three most important features are presented in Table 4 (all models in Table S1 of the supporting material). The full feature importance results of the selected models are provided in Fig. S10.

**Table 4: Random forest (RF) results for predicting the observed groundwater responses including the three most important features from permutation. Performance is given as mean accuracy for classification and coefficient of variation ($R^2$) for regression models across the cross-validation iterations. The number of samples differs between RF models for all wells and only the wells from regional clusters. Note: For RF regression only results with $R^2>0.4$ are shown, all RF models are provided in the supporting material Table S1. Feature importance is given as the mean (across cross-validation iterations) of median importances of the permutation repetitions.**

| RF model | Predicted variable | Number of samples | Performance (accuracy, $R^2$) | Feature | Importance (mean of medians) |
|---|---|---|---|---|---|
| classification | cluster (all) | 6620 | 0.68 | PET_SI | 1.43 |
| | | | | mean_gwdepth | 1.13 |
| | | | | dem | 1.08 |
| | cluster (regional) | 5120 | 0.79 | PET_SI | 1.81 |
| | | | | mean_gwdepth | 1.25 |
| | | | | dem | 1.20 |
| regression | respt$_{SPEI}$ | 5120 | 0.42 | mean_gwdepth | 1.26 |
| | | | | PET_SI | 1.12 |
| | | | | dem | 1.05 |
| | acc$_{SPEI}$ | 5120 | 0.41 | mean_gwdepth | 1.23 |
| | | | | PET_SI | 1.12 |
| | | | | dem | 1.06 |
| | resid_sen$_{SPEI}$ | 6620 | 0.42 | PET_mm | 1.11 |
| | | | | y18_artificial_10km | 1.09 |
| | | | | PET_SI | 1.06 |
| | resid_sen$_{SPI}$ | 6620 | 0.41 | PET_mm | 1.09 |
| | | | | y18_artificial_10km | 1.08 |
| | | | | PET_SI | 1.07 |

The RF classification model of all eight clusters reached an accuracy of 0.68, with accuracies ranging from 0.22 and 0.51 for the long-term trend clusters lt_dec and lt_inc up to 0.85 for the high-frequency southwestern (sw_hf) cluster. The performance improved to an accuracy of 0.79 when predicting only the six regional clusters and excluding clusters lt_inc and lt_dec. In both models, the most important features were the seasonality in potential evapotranspiration (PET_SI) and the mean depth to groundwater (mean_gwdepth) with higher feature importance values for the 6-cluster model.

In the case of RF regressions, the highest $R^2$ in the models including all wells was 0.42 for the trend in SGI-SPEI$_{acc}$ residuals (resid_sen$_{SPEI}$), followed by SGI-SPI$_{acc}$ residuals (resid_sen$_{SPI}$) with $R^2=0.41$. Both models showed the mean annual potential evapotranspiration (PET_mm) and the fraction of artificial surfaces within a 10km radius (y18_artificial_10km) as the most important features. Similar performances were reached for the models predicting the response (respt$_{SPEI}$) and accumulation (acc$_{SPEI}$) time of the 6-cluster data subset, with mean_gwdepth and PET_SI resulting as the most important features. All other regression models had lower performances ($R^2<0.4$) and thus feature importance is not discussed further, although there is high overlap in rankings (Table S1).

The most important feature distinguishing the clusters in the RF models from each other is the seasonality in evapotranspiration (PET_SI). This meteorological spatial feature differs for the different regions, particularly, cluster sw_lf and sw_hf prevalent

in Southern Germany have lower PET_SI values, whereas the northeast (esp. ne_lf) has the highest PET_SI values (Fig. 4 panel a, Fig. S9). RF predictions reflect these differences as shown in the PDP plots for the 6-cluster model (Fig. 4 panel a).

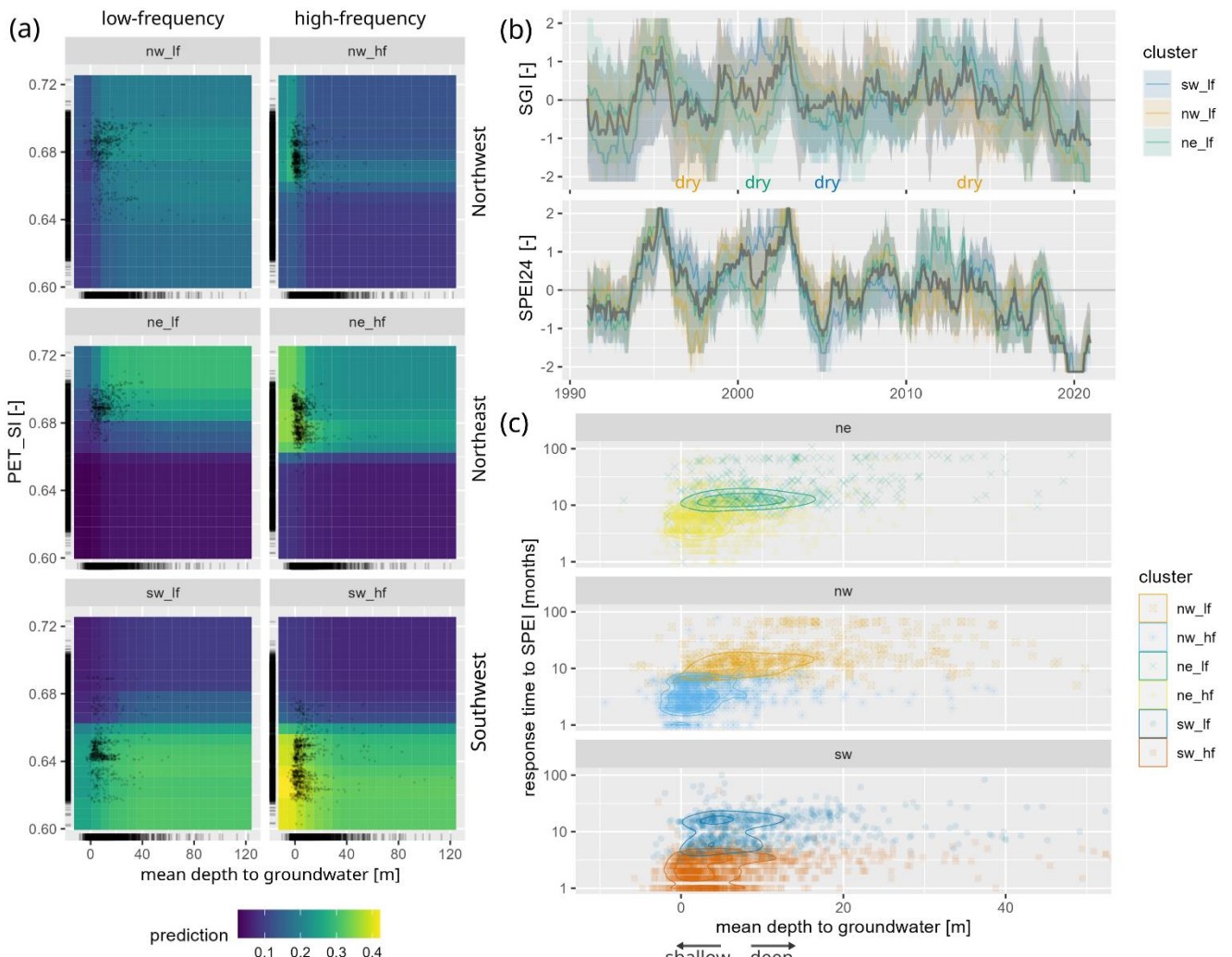

**Figure 4: Meteorological and landscape controls of observed groundwater response patterns: (a) 2D partial dependence plot (PDP) of the effects of mean depth to groundwater (mean_gwdepth) and seasonality in evapotranspiration (PET_SI) on the predicted probabilities of the 6-cluster RF classification model, (b) time series of the mean SGI and SPEI$_{24}$ of all (gray) and of the regional low-frequency (lf) clusters (colored lines) and the 5th-95th percentile range as shaded areas, (c) response time to precipitation-evapotranspiration anomaly (SPEI) respt$_{SPEI}$ versus mean depth to groundwater for cluster members of the six regional clusters with 2D kernel density estimates for probabilities 0.05 and 0.1. Note that mean_gwdepth can be negative in some cases due to data uncertainty from the approximation method using a DEM or in case of artesian groundwater conditions.**

Apart from the control-response relationships learned by the RF models, comparing the SGI and meteorological time series reveals that groundwater anomalies vary more across locations than those in the meteorological drivers. This is shown by the

different bandwidths representing the spatial variability of the SGI versus the $SPEI_{12}$ across all wells (Fig. S6) and the $SPEI_{24}$ across wells of the low-frequency clusters only (Fig. 4 panel b). The latter also shows that main differences between the SGI

time series of the slower responding clusters (nw_lf, ne_lf, sw_lf) are also apparent in the mean regional meteorological anomalies (Fig. 4 b). Examples are the drought in 1997 which was more pronounced in the northwest, the dry period in 2003 in the northeast, while the southwest was rather wet, and the wetter period in the northeast in 2012. This shows that regional differences in the anomalies of the meteorological drivers transfer into the groundwater, thus resulting in distinct groundwater response clusters for different regions.

The second most important feature to distinguish the clusters in the RF models and the most important for metrics of groundwater response times (respt and acc) was the mean depth to groundwater (mean_gwdepth). Here, clusters with a higher frequency in their internal SGI changes (cluster nw_hf, ne_hf, sw_hf) compared to their regional counterparts overall linked to smaller mean groundwater depths below surface (i.e. shallower groundwater). This is apparent in higher sample density of the high-frequency clusters at lower mean_gwdepth values and is also reflected in higher predicted probabilities by RF models

as shown in the partial dependence plots (PDPs) of the 6-cluster RF model (Fig. 4 panel a). Similarly, the tendency of higher depth to groundwater linking to higher response times ($respt_{SPEI}$) within regions is apparent in the data (Fig. 4c) and also reflected in the relationships learned by the RF regression model (see PDP in Fig. S11).

The elevation (dem) was identified as the third most important predictor in the RF classification and regression models of groundwater response times $respt_{SPEI}$ and $acc_{SPEI}$. For the elevation, the differences between low- and high-frequency (lf and

hf) clusters varied between regions: in the northwest the hf cluster was located at lower elevations on average, whereas in the northeast and southwest the hf clusters were located in higher elevations compared to the respective lf cluster (Fig. S9). Additionally, we found that the hf clusters tend to be closer to the streams compared to their regional lf counterparts (Fig. S9). The distance to streams of fourth order (sd_order_4) was ranked as the sixth most important and significant feature (whole range of importances >1; see Fig. S10 panel a and b), followed by the second order stream distance (sd_order_2) in the $respt_{SPEI}$

and $acc_{SPEI}$ RF models. Similarly, the hf clusters more often intersected riparian zones outlined in EEA (2021) with at least 19.6% (nw_hf) and a maximum of 59.8% (sw_hf), while the lf clusters varied between 4.5% (nw_lf) and 31.8% (sw_lf) only.

The RF regression models for predicting the trends in the residuals between SGI and the meteorological anomalies (resid_sen) for all wells include land cover characteristics as a dominant feature, namely the fraction of artificial surfaces within a 10km radius (y18_artificial_10km). Here, a higher urbanization links to more positive trends in the residuals, predominant in cluster

with long-term increasing SGI lt_inc (Fig. 5 panel a, and b). The southwestern low-frequency cluster sw_lt also shows a tendency towards higher $resid\_sen_{SPEI}$ and fraction of artificial surfaces. Cluster lt_inc was additionally more often located in proximity to mining areas as compared to other clusters (Fig. 5 panel c). Thus, cluster lt_inc was found to be overall linked to higher urbanization levels and mining areas.

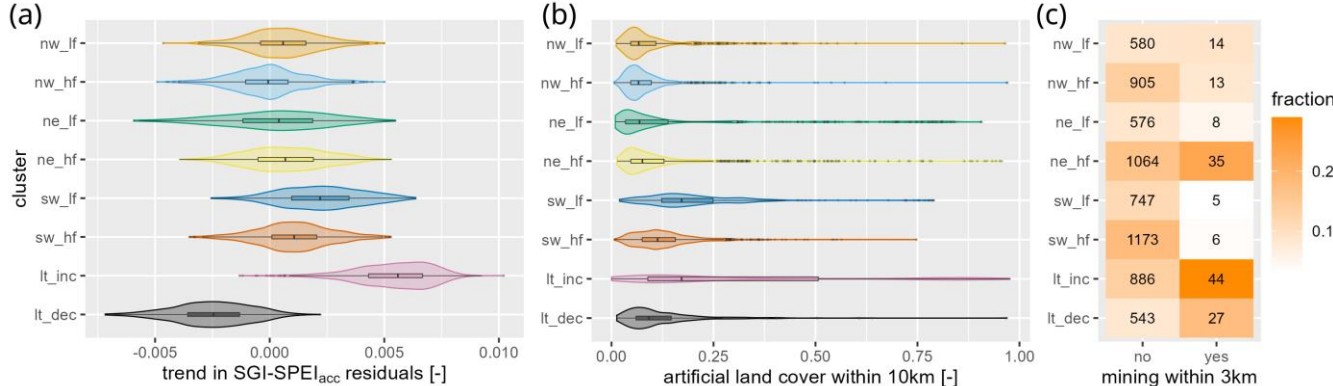

**Figure 5: Trends beyond meteorological drivers link to anthropogenic controls, particularly the long-term increasing SGI cluster lt_inc has higher fraction of artificial land cover and proximity to mining areas than other clusters: (a) cluster-wise distributions of the trend in the residuals of the SGI with the identified SPEI$_{acc}$ (resid_sen$_{SPEI}$), (b) cluster-wise distributions of the fraction of artificial land cover class within a 10km distance based on the CORINE land cover map map from 2018 (EEA, 2019b), (c) the number of wells in mining-proximity (3km) per cluster and heatmap color according to the fraction of cluster within the group of proximity (yes) or no proximity (no).**

### 3.4. Vulnerability of groundwater to meteorological droughts

The vulnerability of groundwater systems was classified into vulnerability to short, medium and long-term meteorological anomalies based on percentiles of the respt$_{SPEI}$ characteristic of the six regional clusters (Fig. 6). The class of short-term vulnerability has response times up to 3.5 months (containing 35.2% of wells), whereas the class of long-term vulnerability responds only after more than 9 months (31.8%). Note that the clusters with long-term trends overlaying the meteorological controls (lt_inc, lt_dec) were excluded from this assessment, as the response time metrics cannot be considered representative of the climatic-groundwater system response for these two clusters.

The spatial pattern of vulnerabilities shows a high variability within regions, reflecting the individual response time scales and the concurrent occurrence of both fast- (hf) and slow- (lf) responding clusters within regions. Nevertheless, the represented northeastern groundwater wells have a slight tendency towards medium or long-term vulnerabilities, as particularly the faster responding cluster (ne_hf) tends to have higher response times with a median of 6 months, compared to 2.5 (sw_hf) and 3.5 (nw_hf) months for the other hf clusters (Table 3, Fig. S8). This reflects the slightly higher acc$_{SPEI}$ values within ne_hf (Fig. 3). Accordingly, across regions, we found the highest share (45.4%) of long memories (respt$_{SPEI}$>9 months) in the northeastern clusters (ne_lf, ne_hf). About half (49.9%) of the wells with short response times (respt$_{SPEI}$<3.5 months) were allocated to the southern clusters (sw_lf, sw_hf).

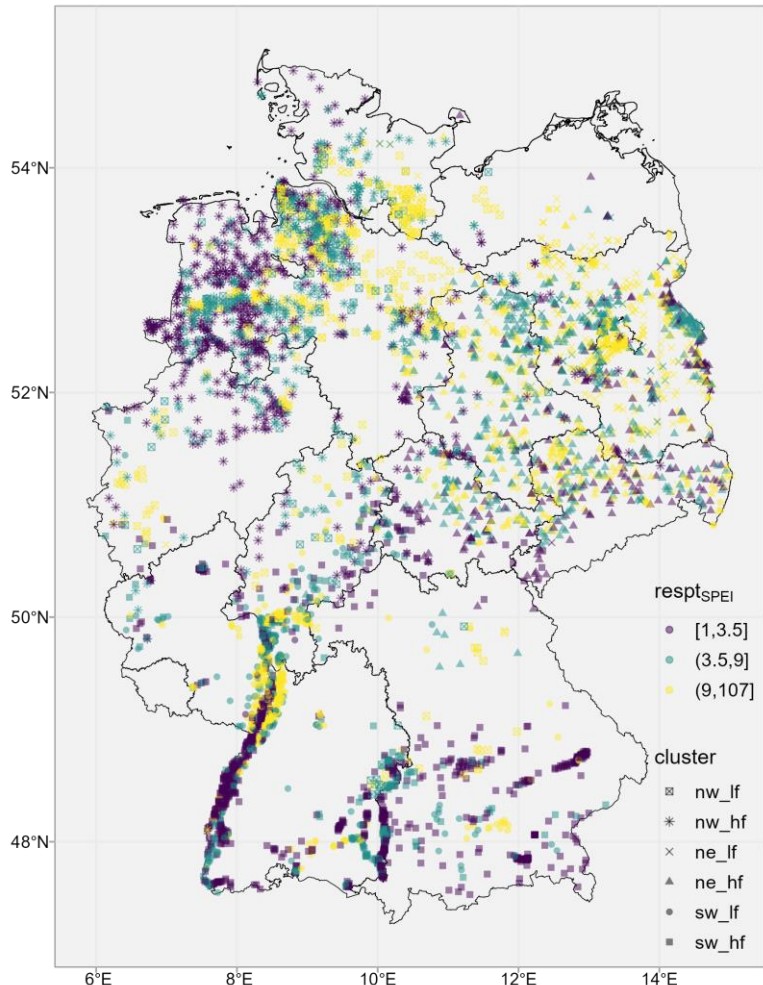

**Figure 6: Vulnerability classes of groundwater systems to short- (purple), medium- (green) and long-term (yellow) meteorological anomalies based on the 33$^{rd}$ and 67$^{th}$ percentiles of the response time of groundwater anomalies SGI to the precipitation-evapotranspiration anomaly SPEI (respt$_{SPEI}$ in months) from the member of the the six regional clusters (Table 3).**

## 4. Discussion

### 4.1. Spatial variability in groundwater responses

Out of the eight identified clusters, six, in three pairs of two, were predominant in three distinct regions, while two clusters were distributed across Germany (Fig. 2). Overall, the spatial variability in groundwater head anomalies was found to be larger than that in the meteorological driving forces (Fig. 4 b, Fig. S6). This is not surprising, as a relatively high similarity in meteorology arises from spatial coherence in the occurrence of meteorological extremes like precipitation deficits and

temperature anomalies resulting from stable atmospheric conditions across large scales (Hari et al., 2020; e.g., Christian et al., 2023). This contrasts with a high variability in hydrological processes in the subsurface and resulting site-specific groundwater dynamics (e.g., Heudorfer et al., 2019; Lischeid et al., 2021; Kumar et al., 2016). This applies even though the spatial extent, duration and severity of meteorological droughts strongly vary across drought events (Oikonomou et al., 2020; Rakovec et al., 2022). Similar observations have been described, for example, by Bloomfield and Marchant (2013) and Kumar et al. (2016).

We found a majority of negative trends in the SGI time series and a high number of minimum SGI values in the last years, which is in line with prevalent negative monotonic trends and minimum heads described by CORRECTIV (Donheiser, 2022). Negative trends in the regional clusters were more predominant in northwest Germany, followed by the northeast with still mostly negative trends and the southwest with a balanced distribution of trends (Fig. S8, sen slope SGI). These mostly reflected trends in meteorological drivers, in contrast to the trends of the clusters with dominant long-term trends (lt_inc, lt_dec), which deviated from those of the drivers (Fig. 5a). The SPEI shows higher negative trends than the SPI (Fig. S5) in response to the increase in temperature and thus potential evapotranspiration with global warming during the study period including the exceptionally hot summers in 2018 and 2019 (Vicente-Serrano et al., 2010; Hari et al., 2020), in line e.g. with differences between SPI and SPEI trends in Europe found by Ionita and Nagavciuc (2021). This explains the higher correlations between cluster lt_dec with SPEI time series compared to lt_inc and vice versa for the SPI.

The groundwater response characteristics differed across clusters, with three clusters showing a prevalence of high-frequency in the SGI variability representing fast responses and shorter system memories (nw_hf, ne_hf, sw_hf), whereas another three clusters exhibited lower frequencies in SGI variability representing slower responses and longer system memories caused by dampening and attenuation of the variability (nw_lf, ne_lf, sw_lf; Fig. 3). The median accumulation time (as a measure of memory time scale) across the 6,626 wells of 13 months was similar to previous studies (Bloomfield and Marchant, 2013; Kumar et al., 2016). The differences in system memories were closely linked to the groundwater drought characteristics of the clusters, with systems with shorter memories experiencing shorter and less severe groundwater droughts (in terms of accumulated SGI), but facing drought events more often (Fig. S8), in line with previous studies (Bloomfield and Marchant, 2013; e.g., Bloomfield et al., 2015). The overall high variability in optimal accumulation times underlines the finding by Kumar et al. (2016) that groundwater droughts cannot be described by a uniform meteorological drought index (in the form of SPI with one accumulation time) and corroborated this finding also in terms of the SPEI and for Germany as a whole.

Both the autocorrelation length and the optimal accumulation time identified with cross-correlations can be considered metrics of system memory. Accordingly, we found both to be lower on average in the identified fast-responding (hf) clusters and higher in the slow-responding, low-frequency (lf) clusters. Bloomfield and Marchant (2013) found the two metrics autocorrelation length and SPI accumulation period to align with a correlation coefficient of 0.79 across 14 groundwater wells in the UK. However, across the large sample (6,626 wells) in our study, this relationship could not be confirmed with the same

strength, i.e. $r_{spearman}=0.64$ for $acc_{SPI}$ and $r_{spearman}=0.60$ for $acc_{SPEI}$. This questions the transferability of one metric to the other and the generality of this link at the level of individual wells. Although both metrics represent memory time scales and are

465 related, they ultimately describe different properties: the optimal accumulation time represents the system's memory for past meteorological drivers, while the autocorrelation length represents the overall persistence in the time series resulting from the sum of effects on groundwater dynamics. Nonetheless, both metrics can be affected by interacting effects. For example, some deviations might be caused by the superposition of long-term trends that interfere with the identification of these intrinsic system response properties. Bloomfield et al. (2015) e.g. described weaker cross-correlations between SPI and SGI time series

in the case of significant trends in the SGI. Interestingly, the SPEI accumulation times and autocorrelation lengths deviate most for the clusters with dominating long-term trends and, although less, for the high-frequency (hf) clusters and the northeastern low-frequency (ne_lf) cluster (Fig. S7). This could indicate that those systems are less strongly linked to meteorological drivers. Although, relatively high correlation coefficients with the SGI from cross-correlation ($cc_{SPI}$ and $cc_{SPEI}$ respectively; except for the cluster with long-term increasing SGI lt_inc) do not generally support this interpretation. Another factor could

be the method of identifying accumulation times based on the absolute maximum in correlation coefficients, while similar values could be reached for several different accumulation times in the case of stagnating accumulation, as observed, for example, for the cluster with long-term decreasing SGI (lt_dec) regarding the SPEI accumulation times (Fig. 3a).

## 4.2. Controls of groundwater response dynamics

We could demonstrate that the interplay between the meteorological drivers, landscape filtering and anthropogenic impacts

control groundwater response patterns and separates them into clusters as discussed based on the RF model results in the following. This means that different controls jointly operate to cause distinct groundwater head responses at individual locations. This is also supported by the fact that similar features ranked high in the different RF models e.g. the 8-cluster and 6-regional-cluster classification model, although with varying performances and feature importance (Table 4). The similar rankings provide confidence in the robustness of results even if model performances are not high in the regression models with

$R^2 < 0.5$. This range of performance is, however, not surprising given the heterogeneity of subsurface conditions and complexity of processes, which cannot be fully represented by the simple characteristics used as predictors. The highest model performances of a mean accuracy of 0.79 for predicting the six regional clusters and 0.42 for the groundwater response times regarding the $SPEI_{acc}$ out of the different RF models are comparable to that of Schuler et al. (2022). They reached a model performance ($R^2=0.49$) in the out-of-bag evaluation of RF models for predicting autocorrelation lengths at 114 wells in Ireland.

**4.2.1. Different responses across regions link to meteorological drivers**

Meteorological drivers were identified as the major control for distinguishing groundwater head anomalies across regions based on the RF results and the regionally temporal coherence in $SPEI_{24}$ and SGI time series (Fig. 4). Three main regions with predominant clusters were identified, i.e. Northwest, Northeast, and Southern Germany.

On average, the meteorological drivers (SPEI, SPI) could explain 50% of the temporal variability in groundwater heads (corresponding to the median of cross-correlation coefficients of r=0.7). This is in the same range as in previous SGI investigations (Kumar et al., 2016; Bloomfield et al., 2015). This high explained variance was predominant in the six regional clusters, whereas the national clusters with dominant long-term trends (lt_inc and lt_dec) were less strongly cross-correlated with the SPEI and SPI respectively (Fig. 3, Fig. S8).

We argue that meteorological anomalies control interregional differences in groundwater head anomalies. As shown for clusters with longer response times (Fig. 4 panel b), major temporal differences in the regional meteorological anomalies are reflected in the average groundwater anomalies of the same regions. For example, drier periods in the northwest occurred around the years 1997 and 2004, as well as wetter periods in the northeast around 2012 and in the southwest around 2001. In recent years (around 2018-2020), Germany as a whole faced a severe meteorological multi-annual drought (e.g., Rakovec et al., 2022) with similar average meteorological anomalies (based on $SPEI_{24}$) mostly reaching their absolute minima across the 30 years, translating into wide-spread severe groundwater droughts across Germany. Regarding the groundwater clusters with longer response times (lf clusters), the two northern clusters (nw_lf, ne_lf) faced more severe groundwater droughts (reflected in more severe anamolies, i.e. lower SGI values) on average in these recent years compared to the southern cluster (sw_lf). For the Netherlands, the spatio-temporal development and recovery of the groundwater drought was connected with regionally different courses of the meteorological drought severity during 2018-2020 (Brakkee et al., 2022), which highlights the regional control of the meteorological driver.

The feature importance results from the RF classification models for predicting the clusters underline this observation. In both classification models, the seasonality in evapotranspiration PET_SI turned out to be the most important predictor. Nevertheless, in the 6-cluster model, the model performance (accuracy=0.79) and feature importance of PET_SI (1.81, i.e. prediction error reduced by more than 80%) were higher than in the 8-cluster RF model. This indicates that the predictive power of PET_SI relates to the regional differentiation of the clusters rather than the two long-term trend clusters. PET_SI varies dominantly across regions with a general gradient from southwest to northeast due to the variations in temperature and solar radiation depending on latitude and proximity to the sea (i.e. continental versus maritime climate), leading to the highest seasonal variations in northeastern and northern and lowest values in southern Germany. Accordingly, it has a well-defined regional gradient, which proved to be able to distinguish major regional differences in the drivers and resulting response patterns.

**4.2.2. Different responses within regions link to landscape filtering**

Even though a region is subject to similar meteorological forcing (in terms of anomalies), we found a large variety of groundwater responses within regions. These different responses were mainly characterized by different response time scales, i.e. the frequency in change of the SGI or in other words the system memories, and closely linked to the number, duration and

severity of droughts. Within the three identified regions, landscape filtering (i.e. modulations of the driver signal by the landscape) was identified as the main control of the groundwater response time scale.

The mean depth to groundwater (mean_gwdepth) was found to be the second most important feature in the RF classifications and the highest-ranked feature in the RF regressions for the response time (respt$_{SPEI}$) and optimal accumulation time (acc$_{SPEI}$) of the SPEI (Table 4). The depth to groundwater can be referred to as the thickness of the unsaturated zone in the case of an unconfined aquifer and the depth of the water pressure head below the surface in the case of a confined aquifer. The unsaturated zone or depth to groundwater has been discussed as a major control of memory effects and groundwater dynamics in previous studies (Bloomfield and Marchant, 2013; Haaf et al., 2020; Haaf et al., 2023; Kumar et al., 2016; Lischeid et al., 2021; Wossenyeleh et al., 2020). Mechanistically, this can be explained by the delay of water transport from precipitation to groundwater recharge with long flow paths and water travel times through the vadose zone and by the attenuation as the infiltration front widens due to different flow paths and flow velocities through the unsaturated pore spaces. Wossenyeleh et al. (2020), for example, showed that the groundwater recharge delay is closely linked to the depth to groundwater and can be up to more than four years in Belgium by modelling the flow through the unsaturated zone. Schreiner-McGraw and Ajami (2021), in addition, discussed that the depth to groundwater indirectly controls response times of groundwater to multi-year meteorological droughts, as it covaries with aquifer transmissivity and shifts the relative importance between mechanisms, e.g. between climatic versus geographic controls. Shallow unconfined aquifers can be recharged by relatively fast percolation of local precipitation through a shallow unsaturated zone in addition to distant recharge and topographic convergence of lateral flows. In contrast, in deeper aquifers, local percolation may take significantly longer to reach the water table (Lischeid et al 2021), creating a delayed, attenuated response in groundwater heads. However, in confined aquifers, recharge from more distant recharge zones (e.g. a nearby mountain front) could still produce a more immediate head response via pressure transmission. This underlines the multi-faceted role of subsurface geologic structure and geometry in defining recharge flow paths and in turn depth to groundwater and groundwater head response time scales. Indeed, although mean_gwdepth clearly turned out to be the most important landscape predictor in the RF models and showed a clear tendency of shallower groundwater linking to shorter response times (Fig. 4), there is scatter around this relationship (Fig. 4 panel c) resulting from interactions with other spatial controls and reflected in explained variances below 50% ($R^2 < 0.5$).

Additionally, response time scales are linked to the topography. Different linkages between elevation and slow, dampened versus fast responses (represented in the lf and hf clusters) in the three distinct regions suggest different mechanisms connecting elevation to response times. On the one hand, higher elevations are usually linked to deeper unsaturated zones (i.e. mean_gwdepth; Haitjema and Mitchell-Bruker, 2005) and thus higher response times, while groundwater heads at lower elevations close to streams fluctuate at shorter time scales (e.g., Peters et al., 2006, Haaf et al., 2023). This could be a dominant process in the northwest of Germany where the fast-responding systems (nw_hf) tend to be located at lower elevations compared to the slow-responding systems (nw_lf; Fig. S9). Similarly, Brakkee et al. (2022) found groundwater response times

to be longer in more elevated areas of the Netherlands coinciding with deeper water table depths. On the other hand, high elevations can be linked to small depths to bedrock and aquifer thickness and thus shorter response times and memories, as e.g. described for Ireland with depths to bedrock below 10m by Schuler et al. (2022). Furthermore, in higher elevation mountainous regions wells often tend to be placed in alluvial valley fills near streams with shallow depth to groundwater and

faster response times in turn. The regional cluster-pairs in the northeast and southwest have smaller overlap in their spatial extent, e.g. in the south the fast-responding hf cluster includes more wells in the Upper Rhine Plain and southeastern regions compared to the respective slow-responding lf cluster. In the northeast, the hf cluster extends more towards the south into more mountainous areas, e.g. the Ore mountains, while the lf cluster is centered more in Brandenburg. Note the positive (although not strong) correlation between the mean_gwdepth and topographic variables: for the elevation r=0.20, for the slope r=0.43 or

the twi r=-0.38 (Spearman correlation, Fig. S8). High mean_gwdepth was linked to higher topographic slopes and lower wetness indices, while the link was less clear for absolute elevation, because of the higher relevance of relative height in the hydrologic system between the water divide and stream network (Schuler et al., 2022; Haaf et al., 2020; Rinderer et al., 2017; Haitjema and Mitchell-Bruker, 2005). This could explain the importance of elevation in the RF models, which can represent nonlinear relationships, while the underlying processes cannot be uniquely interpreted across regions.

Closely linked to the topography, the RF models further indicated a link between clusters and response times to the distance to stream. The hf clusters with shorter response times and memories (i.e. SGI changes at a higher frequency) tended to be located closer to streams than their regional counterparts with longer memories (lf clusters, Fig. S9). This is in line with Peters et al. (2006), who found higher attenuation of groundwater drought signals closer to the water divide than closer to the stream. Similarly, Haaf et al. (2023) showed that overall locations closer to streams tend to show higher flashiness in daily groundwater

heads and pointed out the nonlinearity in the controls with higher importance of stream distance during wet conditions. In proximity to streams, groundwater dynamics are typically directly linked to interactions between groundwater and surface waters (Haaf et al., 2023; Nogueira et al., 2021). For example, near-stream groundwater heads often respond quickly to stream water level fluctuations via a pressure response and show very similar variability due to the confined or semi-confined conditions commonly found in alluvial aquifers (Bartsch et al., 2014; Gianni et al., 2016). The distance of wells to 4th order

streams (or higher) in our study also varied systematically between southwestern and northern regions, which likely results from the proximity of most wells to the Rhine (order 7) in the Southwest. For larger stream orders, Belitz et al. (2019) also pointed out that the distances are more descriptive of overall location than process, in contrast to smaller stream orders, where the distance to stream links more directly to hydrological mechanisms. For these small orders Belitz et al. (2019) showed a generally positive relationship between the well locations relative to the stream and the water divide and the water table depths

based on random forest models. In our study, this link between distance to stream and groundwater depth was however weak across the wells, i.e. the highest correlation between mean_gwdepth and the distances to different stream orders was only r=0.2

for the stream distances including the first order (river_dist_m, Spearman correlation, Fig. S4; and r=0.16 for sd_order_1; note: sd_order_1 was not used in RF to reduce redundancies, see Table 1).

The hydrogeological setting, as a landscape property controlling the water flow in the subsurface, has been identified as a
dominant control on groundwater dynamics in several studies. Hellwig and Stahl (2018) found hydrogeological conditions to control response times of baseflow from headwater catchments, which were shorter in fractured aquifers compared to porous aquifers. Several other studies discuss a dominant effect of aquifer transmissivity, effective porosity, storativity or aquifer thickness on groundwater dynamics (Bloomfield and Marchant, 2013; Haaf et al., 2023; Schuler et al., 2022). In our study, we used only saturated hydraulic conductivity, which we extracted from a hydrogeological map representing the upper aquifers
(BGR and SGD, 2016), because of a lack of data on subsurface characteristics in the used groundwater data set, including information on the aquifer that the wells tap into. There was no clear relationship between the saturated hydraulic conductivity (characterized by kf_rank, Table 1) and response times and cluster, in line with Kumar et al. (2016). Differences in the effects of hydrogeological controls on head response times identified in these studies could be due to several reasons: the overrepresentation of specific hydrogeologic characteristics in the data set (e.g. highly productive aquifers), a
misrepresentation of local hydrogeological conditions at a well location in a coarse hydrogeological map (esp. when local borehole data are missing), or differences in the investigated response variables (e.g. groundwater heads with strong seasonal variations (e.g., Haaf et al., 2020), head anomalies (e.g., Bloomfield et al., 2015), or groundwater discharge as baseflow (e.g., Hellwig and Stahl, 2018)). For example, Haaf et al. (2020) found different controls of groundwater head dynamics for confined and unconfined aquifers for Southern Germany. Also, different effects of controls at local or spatially integrated scales are
likely, as they represent different system characteristics and have been shown to not be directly transferable (Kumar et al., 2016; Hellwig et al., 2020; Van Loon et al., 2017). Studies representing spatially integrated response signals (e.g., raster or catchment integrated indicators such as the baseflow) seem to find a higher relevance of hydrogeological conditions (Hellwig and Stahl, 2018; Hellwig et al., 2020).

In summary, the identified and discussed landscape controls suggest that the spatial variability in local groundwater drought
response time scales (i.e. system memories) within meteorologically distinct regions is dominantly controlled by vertical low-pass filtering through the unsaturated zone and secondarily by controls affecting the lateral flow conditions linking to subsurface hydraulic and storage conditions. At integral landscape (or catchment) scale, the hydrogeological controls of storage and discharge seem to be a more dominant driver of drought propagation time scales. In this study, we did not find a dominant and clear influence of the one hydrogeological variable at hand (saturated hydraulic conductivity), however, additional
hydrogeologic information lacking in our data set, such as the depth of the well screen, aquifer type (unconfined versus confined), aquifer transmissivity and storativity could provide further insights into the causes of variable head responses.

Although differences in response times within regions were found to be larger than between regions in Germany and dominantly controlled by landscape filtering, systematic regional differences in groundwater response times may also be linked to general climatic conditions (humid vs. drier) and related groundwater recharge rates (Berghuijs et al., 2024). The feature importance results from the RF models for $respt_{SPEI}$ and $acc_{SPEI}$ also ranked climatic variables high, i.e. the seasonality in potential evapotranspiration (PET_SI) as 2nd, mean annual precipitation (P_mm) as 4th and aridity (AI) as 5th ranked feature. Cuthbert et al. (2019) showed that arid areas across the globe with low recharge have much longer groundwater response times (i.e. hydraulic memories), which they defined as the time to re-equilibrate when recharge conditions change. In line, Schreiner-McGraw and Ajami (2021) demonstrated that locations with low recharge rates (<200mm y$^{-1}$) commonly experienced slower recovery (recovery times >3 years in half of the wells) from multi-year droughts compared to areas with higher recharge rates. This could be one reason for the overall slightly higher response times of the groundwater systems (memories) in the less humid northeast of Germany with lower average groundwater recharge rates than in the more humid northwestern and southern parts. Especially for the fast-responding clusters, the northeastern ne_hf has longer response times compared to nw_hf and sw_hf (Fig. 3b, Table 3). Another effect could arise from regional differences in landscape genesis. Aquifers across northern Germany were formed by thick, glaciofluvial deposits with stacked aquifers, interbedded with layers of finer, less conductive sediments such as tills and clays (e.g., Lischeid et al., 2021). In contrast, in the South the hydrogeological setting is more diverse including fractured and karstic aquifers as well as alluvial aquifers in river valleys, where many of the wells are located. The latter were formed under periglacial conditions from coarse, conductive sediments derived from the Alps.

### 4.2.3. Anthropogenic impacts cause superimposing trends

Across Germany, two clusters were clearly characterized by long-term trends in the groundwater head anomalies (SGI) and also in the residuals between the SGI and the meteorological drivers (SPI or SPEI respectively). More specifically, they showed dominant increasing (cluster lt_inc) and decreasing (cluster lt_dec) trends that clearly deviated from trends present in the meteorological drivers (Fig. 5 panel a). Those trends are presumably caused by anthropogenic activities, as indicated by the linkages of the cluster with rising trends (lt_inc) and the trend variable $resid\_sen_{SPEI}$ to artificial (urban) and mining areas. The underlying mechanisms are discussed in the following.

Upward trends in groundwater heads (cluster lt_inc) were more prevalent in regions with mining activities, such as the open-pit lignite mining areas in Western and central East Germany, and urban areas including the metropolitan area of Berlin. This suggests that human activities, such as changes in water management are the main cause. Open-pit mining is commonly associated with significant groundwater pumping to keep the mining pits dry, resulting in massive head drops of up to several hundred meters. Observations of rising groundwater heads can thus be linked to decreased groundwater pumping due to the relocation or closure of open-pit lignite mines and may occur in relative proximity to falling groundwater heads. For example, the trend clusters lt_inc and lt_dec both occurred in the Rhenish lignite area (Ger.: Rheinisches Braunkohlegebiet) in central

western Germany (west of Cologne). Many of the lignite mines in the Central German lignite area (Ger.: Mitteldeutsches Braunkohlerevier) were closed in the early 1990s after the German reunification, in line with the positive trends in groundwater heads (lt_inc) prevailing in our data. These effects can be expected to continue, as lignite mining activities are phased out, because of the Coal Exit Act to reduce $CO_2$ emissions in Germany.

In urban areas, changing groundwater heads can be linked to changes in water use. Potential causes include changes in the water demand due to demographic or industrial developments or in the used water sources. Overall, water use in Germany has drastically decreased by more than 50% since the 90s for several reasons including technological improvements (Umweltbundesamt, 2022b). Water demand for energy (mostly cooling water) as the greatest user has strongly decreased, but also the public water use has decreased from 144l $day^{-1}$ $capita^{-1}$ in 1991 to 128l $day^{-1}$ $capita^{-1}$ in 2019 (Umweltbundesamt, 2022a, b). Particularly in Berlin, we found a prevalence of rising groundwater heads (lt_inc cluster), as the water demand, which is mainly met with supply from groundwater in this densely populated area, has decreased by about 42% since the 90s (Umweltatlas Berlin, 2018; Frommen and Moss 2021). This contrasts with observed groundwater head declines associated with large urban areas and tourism and an increased water demand in other temperate regions, such as in parts of France (Chávez et al. 2024). Another reason for changing groundwater heads could be the relocation of resources and supply wells, e.g. due to decaying water quality such as high nitrate or sulfate concentrations, e.g. in Berlin (Marx et al., 2023). It should be noted here that 25% of the wells of the cluster with dominant positive trends in the SGI (lt_inc) are located in Berlin with a very high station density in the data set, esp. in western Berlin, which could bias the identified controls. Nevertheless, the identified effects of changing water demands can be considered transferable to other urban areas, depending on local demographic and industrial settings.

Additionally, active groundwater resources management such as managed aquifer recharge can lead to rising groundwater heads. The region Hessian Ried (Ger.: Hessisches Ried) in the Upper Rhine Plain is a prominent example, where treated Rhine water is infiltrated since 1989 to increase groundwater resources to supply water demand for agriculture and the population of the metropolitan region including the city of Frankfurt (Main) (Staude, 2023; Weber and Mikat, 2011). In line with the subsequent groundwater head increases, our study showed a strong prevalence of the cluster with upward trends (lt_inc) in this region. Similarly, improved groundwater management, including managed aquifer recharge has led to a recovery in groundwater heads in some semiarid Mediterranean aquifers with intense agriculture (Chávez et al. 2024) and other regions such as Arizona, Thailand and Iran (Jasechko et al. 2024). Overall, several local and regional reasons could be identified for increasing groundwater heads, either related to decreased groundwater abstractions or managed artificial recharge of groundwater resources.

Downward trends in SGI not explainable be the meteorological signal alone (cluster lt_dec) could similarly be linked to changes in anthropogenic water use, though no clear spatial controls could be identified. Increased water abstractions can result

from various factors, including demographic change, changes in mining activities or agricultural needs, which can temporarily
be higher during droughts and heat waves, representing a positive feedback loop on water resources. However, hard data on groundwater abstractions are typically hard to get. In our study, the spatial controls associated with such potential increase in water abstractions were either non-unique or data characterizing them was missing in our analysis. This is also reflected in a generally low predictability of the cluster with decreasing SGIs (lt_dec with only 22% correct classifications). Consistent, national-scale data on groundwater abstractions are thus crucial to clearly identify and assess controls and to differentiate
between meteorological and human influences on observed changes in groundwater heads.

Overall, the anthropogenic controls identified in the random forest regression for the trends in residuals between groundwater and meteorological anomalies proved to be more indicative for the positive trend deviations prevalent in cluster lt_inc than for the less predictable negative trend deviations prevalent in cluster lt_dec.

### 4.3. Implications

This study indicated that there is a large spatial variability in groundwater response time scales to meteorological forcing, even within the same region. This implies different vulnerability to the different types of driving meteorological drought events, i.e. meteorological extremes with respect to different time scales represented by different accumulation times.

Systems with short response times, i.e. wells with high frequency of head changes, are more prone to respond heavily to short meteorological anomalies, but can also recover faster when the climatic drivers return to "normal" or wetter conditions.
Extreme short-term anomalies can be particularly critical for stream ecosystems, as members of the high-frequency clusters are more closely connected to streams (Section 3.3 and 4.2.2). In our study, 61.5% of the short-term vulnerable class are located within 500m distance of the nearest stream and 49% within riparian zones (EEA, 2021). The southern clusters represent half of the wells with short response times (respt$_{SPEI}$<3.5 months) and are at the same time more often located in proximity to streams (Fig. 2, Fig. S9), so that regional differences in processes cannot be fully disentangled due to this data bias. Stream
ecosystems and groundwater-dependent ecosystems, such as riparian wetlands, may be severely impacted by short-term droughts if groundwater heads drop, stream discharge falls below the ecological minimum flows, and streams become losing or even fall dry. Moreover, baseflow has been shown to have short response times too, in a range of a few months only (Hellwig et al., 2020). Thus, groundwater systems with short response times seem to imply a high ecological drought vulnerability. In addition, small, fast-responding aquifers, like local riparian aquifers, may be highly susceptible to short-term droughts (Schuler
et al., 2022) and thus may require backup water supply resources.

Systems with long response times, i.e. wells with a low frequency of head changes, show a response when meteorological anomalies accumulate over longer periods, such as more than a year. At the same time, these systems recover only slowly and retain a long memory, potentially leading to legacy effects from past management or climate conditions even after (driving)

conditions normalize. While they can buffer short-term climatic fluctuations and thus serve to bridge short droughts of a few months regarding water demands, they are more vulnerable to extended droughts or overuse due to their long recovery times. Consequently, locations with long response and recovery times may be particularly at risk from consecutive droughts, increasing with climate change (e.g., Rakovec et al., 2022), if the intervals between extreme events are too short or precipitation and subsequent recharge events are insufficient for full recovery of the groundwater storage lost during the drought (Schreiner-McGraw and Ajami, 2021). A large-scale groundwater modelling study for Germany, which assessed recovery times after severe drought under long-term average recharge conditions, found the longest recovery times of more than five years in the Northeast (Hellwig et al. 2021). This is in line with the regional patterns of groundwater response times observed in this study, as the northeastern clusters included almost half of the wells (45.4%) with long memories (respt$_{SPEI}$>9 months) despite the intra-regional variability of observed response times. Identifying and understanding these systems with long-response times might be particularly crucial for water management, and could thus be classified in terms of management-sensitivity.

Different implications for the different response patterns can be derived from changes in the climatic variability due to global warming.

- Firstly, hydroclimatic seasonality is expected to increase, i.e. longer periods of heat waves and precipitation deficits in summer as well as more heavy rainfalls in the wet winter periods (IPCC 2023). This change is reflected in past observations of seasonal meteorological drought indices showing more negative trends in summer SPEI$_3$ compared to other seasons (Ionita and Nagavciuc, 2021). For the future, the more extreme meteorological seasonal conditions are projected to result in increases of winter (main recharge season) and decreases of summer groundwater recharge in Central Europe, although a high uncertainty resulting from different climate projections as well as hydrological models exists (Kumar et al., 2025). This is expected to affect both groundwater head and baseflow seasonality (Hellwig et al., 2021). In this context, Hellwig and Stahl (2018) discussed that catchments with short response times could be more prone to decreasing low flows, as precipitation deficits in summer could not be buffered across seasons in such systems. Wunsch et al. (2024) also found that meteorological conditions during summer mainly control low groundwater heads in fall in shallow unconfined aquifers, which cannot be prevented by a preceding wet winter. However, there has been little reflection on the dependence on response times in groundwater, which we found to be spatially highly variable. Indeed, we also found groundwater wells with subseasonal response times (about one third of wells had response times respt$_{SPEI}$≤3.5 mon). These groundwater systems with short-term memories could thus potentially be more strongly affected by the increasing hydroclimatic seasonality. As these fast-responding systems are often located in proximity to streams (Fig. S9, Section 3.3 and 4.2.2), this could lead to an increase of losing or intermittent streams and further exacerbate ecological drought vulnerability.

- Secondly, multi-year or consecutive meteorological droughts are expected to increase in frequency and intensity with climate warming (e.g., Rakovec et al., 2022; Intergovernmental Panel on Climate Change (IPCC), 2023). They will likely affect the groundwater systems with long memory more heavily as time for recovery in between may be too short. For example, in the recent multi-year drought (2018-2020) 45.9% of the slow-responding (lf) wells experienced their minimum mean annual SGI (since 1991), out of which only <1% had their minimum in 2018, but 58.9% had their minimum in 2020, while meteorological anomalies on an annual basis were strongest in 2018. In contrast, 17.5% of the 41.5% of the fast-responding (hf) wells with minima in 2018-2020 had their minimum already in 2018, and about 49.4% in 2019. Droughts of this severity or more challenge water management, particularly, considering the positive feedback between climatic condition and human use as water demand increases during heatwaves and droughts.

- Lastly, increasing temperatures impact water balance components in multiple ways, such as by increasing potential evaporation and glacier melt as well as decreasing snow-fall in the long term (e.g., Fontrodona Bach et al., 2018). Such changes in the water balance are likely to affect future groundwater resources through changes in groundwater recharge, however, this is out of the scope of this study.

As the derived response times of the cluster members with a dominant long-term trend (lt_inc, lt_dec) are likely not representative for the climatic-groundwater system response, the vulnerability map only refers to the other six regional cluster members. Nevertheless, the clusters with long-term decreasing trend might be specifically vulnerable, not only to variability in the climatic driver, but also to superimposed changes in boundary conditions, such as from anthropogenic activities or long-term changes in the climate.

Inspired by the discussion in Bloomfield et al. (2015), the identified SGI clusters could also be used to identify representative wells for larger regions or aquifers (i.e. the specific clusters) and for short term forecasting using seasonal hydroclimatic forecasts. Both aims could be combined into representative (for cluster) and meteorologically "well-behaved" (high anomaly cross-correlation) wells representing characteristic systems. This would however need further evaluation.

Further, spatially comprehensive predictions of response time scales would be desirable to inform water management which relies on spatially representative information on the groundwater status. However, the fact that the main identified control of response time scales within regions, i.e. mean depth to groundwater, is not known in space and, in addition, can change in the long term challenges this goal. Potentially, for areas where mean groundwater depth can be associated with other controls, such as the topography, spatial predictions could be tested (Schuler et al., 2022). Further research is needed to establish a spatially seamless mapping of groundwater drought vulnerabilities in Germany.

## 5. Conclusions

This large-sample analysis of groundwater head anomalies across 6,626 wells in Germany overall revealed a high spatial variability in groundwater head responses to meteorological anomalies. Within this variability, wells were grouped by similarity in groundwater head anomalies into six regional clusters, distinguished by three meteorologically distinct regions and two response time scales, and two countrywide clusters. The identified regions with similar response patterns were the northwest, northeast and the south of Germany. The median response time scales in terms of meteorological accumulation times ranged from a few months for systems with shorter memory to several years for systems with longer memory (or persistence). The characteristic response time scales were closely linked to the frequency, duration, and severity of groundwater droughts.

The main cause for distinct groundwater responses (represented by the SGI) across regions was found to be the differences in the meteorological driving forces (SPI, SPEI). These drivers could on average yield cross-correlation results of r=0.7 considering the individual optimal response time scales. Variables defining landscape filtering, in particular, the depth to groundwater, were the main controls of response time scales distinguishing the high- and low-frequency clusters within the same regions. Apart from that, long-term trends in the SGI superimposing the meteorological drivers defined the two countrywide clusters and were attributed to changes in anthropogenic impacts. In particular, the long-term increasing trend cluster was linked to urban and mining areas potentially associated with decreased abstractions due to ceased mining (e.g. in the Central German lignite mining area), and declining water use (e.g. in the city of Berlin), or to artificially recharged groundwater (e.g. in Hessian Ried).

The vulnerability of groundwater systems to different meteorological droughts was classified according to their short-, medium- or long-term response times with different implications for ecosystems and water management. Fast-responding systems, prevailing in the proximity of streams, might be at higher risk with increasing seasonality in meteorological drivers under climate change, as hydroclimatic variability cannot be buffered across seasons. Slow-responding systems could be more affected by consecutive and multi-year droughts, as experienced recently in the 2018-2020 drought and projected to increase in frequency and severity under climate change, due to long recovery time scales.

Overall, this study increased the understanding of dynamic groundwater responses to droughts and their different regional and local controls and derived vulnerability classes within Germany. The distinct responses to meteorological drivers reveal different implications to be expected under climate change. These insights can inform policymakers, water resource managers, and stakeholders for developing effective strategies for mitigating the impacts of droughts on groundwater systems and ensuring sustainable water management practices.

## Author contributions

PE designed the study and conducted the research. RK and AM contributed to the development of the methodological design and supervised the work together with JF. RK provided the meteorological data. PE prepared and revised the manuscript with contributions from all co-authors.

## Competing interests

At least one of the authors is member of the editorial board of the journal.

## Data availability

The groundwater head data used in this study is available from the repository (https://github.com/correctiv/grundwasser-data) provided by CORRECTIV (Donheiser, 2022). Additional extracted data is referenced in the manuscript in Sect. 2 and Table 2. The SGI time series for 6626 wells, derived drought characteristics and spatial properties are provided under https://doi.org/10.4211/hs.f0f661457b5e42288378b13ad0951d90 (Ebeling, 2025).

## Acknowledgments

We greatly thank the CORRECTIV.Lokal network for compiling the data set of mean groundwater heads across Germany "Grundwasseratlas" and making it freely available in a repository (https://github.com/correctiv/grundwasser-data). CORRECTIV.Lokal is part of the non-profit editorial team of CORRECTIV and is a network of more than 1000 journalists in Germany. Their Grundwasseratlas is a highly valuable contribution to transparency for the public and open data and knowledge. We recognize these important efforts, as both journalists and scientists often face the same challenge of inconsistent data availability and formats. We thank Rafael Chávez García Silva for an initial exchange of ideas and Sarah Haug for her help in data preprocessing. We additionally thank the two anonymous reviewers and the editor Alberto Guadagnini for their valuable feedback.

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
