# Peer review of "Groundwater head responses to droughts across Germany"

_EGUsphere, 2024_

## Author Comment (AC1)

**Response to Referee #1**

Dear reviewer, we thank you for your assessment of our work and constructive feedback to further improve our manuscript. We address all your comments in detail below.

**R1.C1**

This study employs a large-sample analysis of wells with high spatial variability in groundwater head responses to meteorological anomalies in Germany. These wells were grouped by similarity in groundwater head into six regional clusters, three meteorologically distinct regions, two response time scales, and two countrywide clusters. The results show that the median response time of the meteorological accumulation times ranged from a few months to several years for systems with longer memory, showing a close link to the frequency, duration, and severity of groundwater droughts. This is an important research that did a good job in classifying and analyzing this large data set. While this work approach clearly originates from the "Large-sample data-driven analyses" as clearly stated at the beginning of the paper (Line 57), it still aims to provide insight to hydrologists and policymakers as summarized at the end of the paper: "Overall, this study increased the understanding of dynamic groundwater responses to droughts and their different regional and local controls and derived vulnerability classes within Germany. The distinct responses to meteorological drivers reveal different implications to be expected under climate change. These insights can inform policymakers, water resource managers, and stakeholders for developing effective strategies for mitigating the impacts of droughts on groundwater systems and ensuring sustainable water management practices."

This is highly appreciated as these communities can provide insight into the speculated mechanistic reasonings arising from the data in this work, which can be related to hydrological aspects (groundwater response time relation to droughts climate events) and urban anthropogenic drivers (the decline in wells usage in Berlin). However, this is also the weak part of this study. Currently, the methods and results section is written in a way that mainly caters to the hydro-informatics community, while the Implications, Conclusions, and part of the Discussion section aim to include hydrologists and policymakers, in various levels of success. This is not a style difference, but an approach that heavily relies on the data analysis and categorization of it instead of using this analysis to provide and support the general conclusion arising from the data, a conclusion that is definitely there and is relevant to these communities. In the following, I will provide three bullets that exemplify this:

- **R1.C1.1:** In the paper, there is a constant reference to the cluster instead of referring to their mechanistic interpretation, which prevents the reader from a clear understanding of how the observation is related to a specific mechanism. For example, lines 564-581 deal with why groundwater heads are increasing in urban areas and how this change is apparent in cluster lt_inc. However, why this cluster is indicative of the water level increase is not stated clearly. While it is clear that this is data-driven research coming from a hydro informatics standpoint, the relevance to the hydrology community resides in drawing the mechanistic aspects between markers, like the clusters, to the processes, like the water level, and the reason for the observed change and correlation.
- **R1.C1.2:** The following statement in line 593: "This study indicated that there is a large spatial variability in groundwater response time scales to meteorological forcing, even within the same region. This implies different vulnerability to the different types

of driving meteorological drought events, i.e. meteorological extremes with respect to different time scales represented by different accumulation times." It is indeed important and should be of interest to both the hydrological community and policymakers, yet the source of variability is not clearly presented in the paper in a way that is accessible to these communities. At the moment, it requires meticulous effort to follow the cluster names, indexes, acronyms, etc. The conclusions and implications are indeed important to these communities, making it very suitable to HESS, yet an effort should be made to "talk" to these communities. To do so, add to the names, indexes, and acronyms the hydrological aspects that they represent within each relevant part. This way, readers from this community can easily trace the relevant aspects of their community and appreciate the result's relevance without being bogged down by the details.

- **R1.C1.3:** I find section 4.3 to be extremely important. This section draws its importance from the conclusions driven by the data analysis, yet understanding how these conclusions relate to the data in this work is really hard to deduce. For example, in line 627: "These locations with short-term memories could potentially be more strongly affected by increasing hydroclimatic seasonality. As these systems are often located in proximity to streams, this could moreover result in more losing or intermittent streams, as mentioned above." Where is the role of losing streams in the context of hydroclimatic seasonality mentioned? The closest relation to the data I found was in line 344: "The distance to streams of fourth order (sd_order_4) was ranked as the sixth most important and significant feature (whole range of importances >1; see Fig. S6 panel a and b), followed by the second order stream distance (sd_order_2) in the resptSPEI and accSPEI RF models.", where the SPEI is for the meteorological anomalies. The need to trace each acronym and parameter is cumbersome, and it worsens when the terms vary through the manuscript (hydroclimatic seasonality or meteorological anomalies?). I'm sure this is the standard presentation for hydroinformatic, but this is not true for other fields. I suggest that the terms will be uniform throughout the manuscript. In addition to parameters and model, a two-word description can be explicitly added where needed, like the above sentence. There are more than 30 terms in this study, and tracking the meaning of all of them is next to impossible in a first, second, or even third read. As such, this should be clearer and more approachable for the reader.

That being said, this is still excellent work that is relevant and highly suitable to HESS. The fact that HESS aims to reach a broad audience from various communities only makes a stronger case for this work to be published there, but it should be altered to be more approachable to these communities.

Response to R1.C1 and subcomments:

Thank you very much for the appreciation of our work and the clear and constructive feedback on how to further improve the communication of our conclusions to a broader community. This is truly helpful to further improve our manuscript to clearly convey our message. We will adapt several parts of the manuscript with the aim to reach out to the different communities, details are given below along the three bullet points (subcomments) that you provided for this main comment and further specific comments below.

In order, to increase the readability, clarity and approachability of the text for a broader community and to increase the link between technical results to mechanisms and interpretation, we plan to adopt several changes throughout the manuscript, summarized here:

1. We reduce the use of acronyms by linking to their meaning in many locations throughout Methods, Results and Discussion Chapter.
2. We move the explanation of the cluster acronyms up in the order of the Results section.
3. We add a new Table to summarize the results across clusters and make them accessible and easily findable to the readers
4. We extend several descriptions in the Methods section for transparency and clarity.
5. We make the link between the observed patterns and the hydrologic mechanisms (i.e. processes) more explicit in several locations. However, we would like to point out that our analysis allows us to discuss dominant processes from the identified linkages between patterns (clusters and other variables like the response times) and thus increases process understanding of the variability of groundwater responses. Nevertheless, we have to acknowledge the remaining variability of the individual wells within the clusters cannot be fully disentangled in such a large-sample approach so that clusters cannot be unambiguously linked to these processes. Therefore, we decided to name the clusters after the observed similarity in the patterns in the groundwater responses, not directly after the discussed and potentially ambiguous mechanisms leading to these patterns.

In the manuscript we will further elaborate on the identified mechanisms behind the patterns observed in groundwater responses to droughts. The dominant processes revealed and discussed in the study are:

- Fast responses are associated with shallow groundwater systems, because of only little dampening by the passage through the unsaturated zone. This leads to a more direct response to the meteorological drivers and overall a smaller memory of the system. These systems are also overall located closer to streams providing an additional boundary condition for the water level fluctuations.
- Slow, dampened responses dominate in deeper groundwater because of larger unsaturated zones that act as a low pass filter and thus attenuate the short-term variability in the meteorological drivers leading to a longer memory of past meteorological conditions.
- Regional differences in the temporal development in the meteorological drivers are reflected in groundwater responses across regions, because the meteorology is the main control of groundwater responses.
- The fast responding wells in the Northeast of Germany respond slower than the fast responding systems in other regions, which can be explained by differences in the climate (drier climate) and related smaller groundwater recharge rates and by quaternary glaciofluvial deposits, which are characterized by sands and gravels interbedded with layers of finer, less conductive sediments such as tills and clays.
- Wells that exhibit dominant long-term trends in groundwater heads beyond the meteorological trends, showed lower correlation to meteorological variability and were linked to anthropogenic controls. Especially for the cluster with increasing trends this link to human activities was apparent.
- Rising groundwater levels link to decreased water use in the area of Berlin and Brandenburg.
- Rising groundwater levels that linked to reduced pumping from mining activities that lowered groundwater levels in the past. This is evident, e.g. in areas of ceased lignite mining in central and western Germany (Mitteldeutsches and Rheinisches Braunkohlegebiet).

- Rising groundwater levels in areas of managed groundwater recharge in the Hessisches Ried.

Response R1.C1.1: We will increase the readability of the section by reducing the use of acronyms and add some formulations that help to link the observed patterns (i.e. cluster with trends) to the dominant mechanisms (i.e. change in water use due to several reasons discussed in the section) revealed by the linkages of patterns to characteristics (random forests, correlations). Please also refer to Response to R2.C4.1 which is related.

Response R1.C1.2: Thank you for this comment. We agree and will carefully go through our manuscript adding the meaning of variables to the acronyms in several locations to more explicitly point out their hydrological meaning. We will further adapt formulations to more explicitly indicate the mechanisms. We will also add a Table providing a summary of results across clusters and mechanisms that makes the results and linkages more easily accessible.

Response R1.C1.3: Thank you for this clear example. We apologize for any confusion the formulations might have caused and will improve their clarity. We understand that meteorology generally refers to all processes in the atmosphere, while hydroclimate include water-related processes as well, which of course are closely linked and overlap, but are not exactly the same. Regarding the definitions of anomalies versus seasonality, we will add a sentence to Section 2.2, Section 2.2.1 and Section 2.2.2 to make the anomaly assessment more accessible. As the approach to assess anomalies in groundwater heads compares values for a given month (e.g., August 2015) to the distribution of all observed values for the same month, it effectively eliminates seasonal variability and thus is clearly distinct from hydroclimatic seasonality discussed in Section 4.3. Therein, we discussed that more extreme hydroclimatic seasonality would affect systems with subseasonal response time scales more heavily than those with longer response times scales, which can better buffer increased seasonal variability. The definition of hydroclimatic seasonality discussed therein is given in Line 619 in Section 4.3. Based on the fact that systems with shorter response times are more often located closer to the streams (as we see in our data, e.g. Fig. S5 "distance to nearest stream, "distance to stream order" 2 and 4 and in the text part that you indicated), we first discussed the implied ecological vulnerability (Lines 596-608) and then in a second step potential stronger effects on these stream systems with increased hydroclimatic seasonality (Lines 619-629). We will change the references to the first part of the discussion (which was "as mentioned above") into the Figure S5 showing the distributions of distances to stream across clusters and Sections where this is mentioned.

The following has some specific comments:

- **R1.C2:** In line 85, it is stated that:" The groundwater head data used in this study are monthly mean groundwater head time series (from originally daily to monthly observations), aggregated." It is unclear why monthly if you have daily data. If not all the data is daily and the aggregation allows to have a data set with the same time scale it should be stated here. If it is done to make the analysis easier, please comment on what we lose in terms of temporal resolution.

Response: We will extend the explanation on the data set and its origin as it was misleading, also in relation to your next comment and comment R2.C4.3. The original data collected by CORRECTIV had heterogeneous temporal resolution, partly daily (as described in the methodology in their repository, https://github.com/correctiv/grundwasser-data ). The data provided by the CORRECTIV network in the public repository and thus available to us is

already aggregated to a monthly resolution, while the underlying higher resolved time series were not provided. We will adapt the text accordingly:

*"The groundwater head data used in this study are monthly mean groundwater head time series across Germany provided by journalists of the CORRECTIV.Lokal network for the period from 1990 to 2021 (Donheiser, 2022; Joeres et al., 2022). CORRECTIV is a non-profit network of journalists who collected the groundwater head time series from the different environmental Federal state authorities responsible for groundwater monitoring in order to report about the groundwater conditions during the recent drought years (Joeres et al., 2022). They homogenized the data by aggregating the original observations (heterogeneous, partly daily resolution) to monthly resolution and provide those in a free repository (for details refer to Donheiser, 2022). This implies that we have a consistent monthly time scale for the analysis, at the cost of having less control on the preprocessing of the original data. For the initial selection of stations of our study, we used the 6,677 stations identified by CORRECTIV based on the criteria: having data for at least 95% of the months, showing no shifts in the head time series, and having station coordinates (Donheiser, 2022)."*

- **R1.C3:** Line 86: A reference and explanation for "CORRECTIV" is needed. I understand this is a German network, and it is referred to in the acknowledgments, but as the data originates from them, a more detailed explanation of the data and how it is collected is needed to appreciate the data quality.

Response: Thank you for this comment. We will add more information on the data source and their data processing, please refer also to the response to the previous comment R1.C2

- **R1.C4:** Line 90: Can't the meteorological data be used to "fill" the gaps instead of linearly interpolating it?

Response: For gap filling, we could have indeed used more extensive ways e.g. using the relationships to meteorological time series. However, this would interfere with the subsequently analyzes of exactly these relationships. We thus prefer to use a simple linear interpolation method for gap filling.

- **R1.C5:** Line 96: What are BGR and SGD? Are they references? Acronyms?

Response: They are indeed acronyms for the German geological authorities providing the data products. BGR is the German Federal Institute for Geosciences and Natural Resources and the SGD stands for the Geological Services of the Federal States. We will add the meaning of the acronyms at the location of the respective first mentioning.

- **R1.C6:** Lines 102, 103: Km2 -> Km^2

Response: Thank you for this hint. We will change it to "wells km$^{-2}$"

- **R1.C7:** Line 104: It's hard to understand which data or wells are important and which are irrelevant from the text. Can it be clarified?

Response: The text part indicated provides information on the spatial heterogeneity of data density. It does not imply that certain wells are more or less important.

- **R1.C8:** Line 147: Can you provide an equation for these SPI and SPEI? Generally speaking, each indicator comes from an equation representing a statistical analysis that has transformed into the jargon of a specific community. In an effort to be approachable to more communities, jargon should be reduced or better defined.

Response: We plan to extend the description of the method to calculate the SGI to the manuscript. As the SPI and SPEI are calculated in the same way, we refer to the section on the groundwater anomaly calculations.

- **R1.C9:** Line 165: Is it possible to elaborate on the method in the supplementary?

Response: Thank you for this suggestion, we will add further information with the Figure of the "elbow plot" in the Supplements.

- **R1.C10:** Lines 195-201: What are the advantages of ML that can't be achieved with statistical analysis and Bayesian statistics?

Response: Indeed, several methods exist that can identify linkages between observed patterns and their controls. The advantage of random forest models is that they can handle non-linearity in the relationships, multicollinearity among the descriptors (spatial controls), and do not need prior assumptions e.g. regarding the distribution of the data. Moreover, random forest is computationally efficient and can easily handle large datasets without a high risk of overfitting due to the averaging across several trees and without the need of extensive tuning or prior regularization. The random forest application is thus very flexible and easily applied to any kind of data and does not require prior knowledge. For this reason, it is commonly used for such purposes. Nevertheless, knowing interdependencies of descriptors should be considered when interpreting the feature importance evaluated in the second step (once the models are trained). Although other statistical methods have different strategies to address these issues, random forest easily combines these advantages without need for elaborate tuning and regularization. Please also refer to the related comment R2.C2.1 by reviewer 2.

We will add text on the main advantages of our approach to the text in the manuscript:

*"RFs are particularly well-suited for efficiently handling large datasets, managing collinearity among descriptors through random feature selection, and identifying complex non-linear relationships without a priori assumptions. Moreover, they are robust to outliers and noise due to their ensemble approach averaging across trees."*

- **R1.C11:** Line 254: The last two paragraphs should be presented or refer to a table in the paper. At the moment, the details are hard to follow and rank.

Response: Thank you for this suggestion, we will add a new table providing a summary of the key results (e.g. autocorrelation lengths, drought event lengths, response times, trends) across clusters presented in these two paragraphs and other locations and the connection to cluster names and their characteristics (referring also to comments R2.C4.6, R2.C4.1, R1.C1.1).

- **R1.C12:** Table 3: What is the acceptable $R^2$? Is there a meaning to $R^2<0.5$? I know that in the hydro-informatic communities, this presentation is "standard," but for a hydrologist, an $R^2<0.7$ is already questionable.

Response: This is a good point, expected performances indeed differ across disciplines and the predicted variables, as model performance depends not only on the noise in the signal but also on the used predictors, their representativeness, underlying assumptions etc. Moreover, also the predicted variables, e.g. the response time, are quantifications of underlying processes containing assumptions, e.g. the transfer from meteorological anomalies (considering different accumulation times) into groundwater in a linear manner.

The model performances that were reached here are in a similar range of a comparable study by Schuler et al. 2022 predicting autocorrelation lengths in groundwater levels in Ireland. Moreover, we think they are not surprisingly low as the predictors used are simple metrics/proxies which are not able to represent the whole complexity of subsurface processes. However, the tendency they show is clear and consistent across the different RF models. This provides confidence in the reliability of model results. Please also refer to comment R2.C4.5.

We will add reflections on the model performance to the manuscript:

*"The similar rankings provide confidence in the robustness of results even if model performances are not high in the regression models with $R^2<0.5$. This range of performance is, however, not surprising given the heterogeneity of subsurface conditions and complexity of processes, which cannot be fully represented by the simple characteristics used as predictors."*

- **R1.C13:** Line 489: "Note the positive (although not strong) correlation between the mean gwdepth and topographic variables: for the elevation r=0.20, for the slope r=0.43 or the twir=0.38 (Spearman correlation)." Note where? is it in a figure?

Response: This correlation was so far an additional information given in the text. We now plan to add a Figure with a correlation heatmap to the Supplements to support this statement.

- **R1.C14:** Line 602, 479, 477 : Fig. appears twice.

Response: Thank you, we will remove the redundancy.

- **R1.C15:** Line 314, 315: Add a space between 4 to panel.

Response: We will adapt it.

---

## Author Comment (AC2)

**Response to Referee #2**

**Critical Review of the Study: "Groundwater Head Responses to Droughts Across Germany"**

The study 'Groundwater Head Responses to Droughts Across Germany' by Pia Ebeling et al. (2024) provides an extensive analysis of groundwater level variations in response to meteorological anomalies over the past three decades. Using a data-driven approach, the study successfully classifies regional groundwater responses and highlights the spatial heterogeneity of aquifer dynamics. While this research is a significant contribution to hydroinformatics and water resource management, there are several areas where its accessibility and interpretation could be improved.

Response: Dear reviewer, we thank you for your assessment of our work and constructive feedback to further improve our study's impact. We address all your comments in detail below.

**R2.C1  1. Scientific Significance**

The manuscript represents a substantial contribution to scientific progress within the field of hydrology, particularly in the context of groundwater response to climate variability. Several aspects of the study highlight its scientific impact:

- Innovative Use of Large-Scale Groundwater Monitoring Data: By analyzing groundwater head responses from 6,626 wells across Germany, this study provides an unprecedented spatially comprehensive assessment of groundwater dynamics.

- Application of Machine Learning Techniques: The clustering approach offers a novel perspective on groundwater system behavior, moving beyond traditional statistical approaches to classify aquifer responses.

- Integration of Hydrometeorological Indices: The study effectively combines groundwater data with standardized precipitation and evapotranspiration indices (SPI, SPEI, and SGI), providing a robust framework for assessing groundwater vulnerability.

- Relevance to Climate Change Adaptation: The study identifies long-term trends in groundwater behavior, contributing to the understanding of how future climate extremes may affect groundwater resources.

Overall, the research presents substantial new data and methodological advancements that contribute significantly to the broader field of hydrology and environmental science.

Response: We appreciate your positive assessment of our work and summary of identified highlights.

**R2.C2  2. Scientific Quality**

The scientific approach and applied methods used in this study are largely valid and appropriate for the research objectives. The methodology is well-structured, employing statistical and machine learning techniques to analyze groundwater responses to

meteorological anomalies. However, some aspects could be improved to strengthen the scientific rigor and clarity of the study:

Response: Thank you very much for the overall positive feedback and constructive comments for further improvements which we address below.

**R2.C2.1  2.1. Validity of Methods**

The study effectively integrates large-scale groundwater monitoring data with meteorological variables using machine learning techniques. While this approach provides valuable insights, a comparison with traditional statistical methods (e.g., Bayesian modeling) could enhance the robustness of the findings. Additionally, justifying the selection of specific clustering algorithms and their interpretability in hydrological contexts would improve transparency.

Response: Bayesian statistics are commonly used to quantify uncertainties and is able to integrate prior knowledge. Thus, it is applied to estimate model uncertainties related to parameter estimation (e.g. Yin et al. 2021, https://doi.org/10.1016/j.jhydrol.2021.126682) or for probabilistic forecasting (e.g. for groundwater contamination Yan et al. 2019, https://doi.org/10.1016/j.jhydrol.2019.124160), and also, more recently, Bayesian optimization is applied for hyperparameter tuning of machine learning models to forecast groundwater levels (e.g., Zhu et al. 2025, https://doi.org/10.1016/j.jhydrol.2024.132567). We are not aware of any applications of Bayesian statistics to model distinct groundwater response patterns to hydroclimatic inputs and link those to spatial controls to understand underlying dominant processes. For this purpose, we used two machine learning methods: (1) the kmeans clustering as an unsupervised method to group the time series of groundwater responses into similar groups and (2) random forest models to link the spatial controls to the observed characteristics.

(1) The kmeans clustering approach groups individuals (here groundwater wells) based on their (dis)similarity in an unsupervised form, i.e. no prior knowledge and assumptions are required. This dissimilarity between two time series is quantified based on Euclidean distance, calculated as the square root of the sum of the squared differences between the corresponding SGI values at each time step. This implies that time series that are fully aligned in their dynamics (and exhibit a similar amplitude) are more similar than, for example, temporarily shifted time series. As Euclidean distance takes squared differences, it is sensitive to noise and single extreme differences. However, it is efficient and thus well suited for large-sample data sets and in particular applicable for time series of the same length.

We will add this information in the corresponding method Section 2.3:
*"The Euclidean distance measures (dis-)similarity based on the squared differences of two SGI time series, making it sensitive to extreme differences and temporal shifts but also computationally efficient."*

(2) The random forest models were used to identify controls of observed characteristics due several advantages regarding data structures and efficiency. Please, refer to our response to R1.C10 from reviewer 1 for more details.

We will add reasons for the use of the RF approach in the manuscript. However, further methodological comparisons like Bayesian modeling although interesting are beyond the scope of this study.

*"RFs are particularly well-suited for efficiently handling large datasets, managing collinearity among descriptors through random feature selection, and identifying complex non-linear relationships without a priori assumptions. Moreover, they are robust to outliers and noise due to their ensemble approach averaging across trees."*

**R2.C2.2  2.2. Discussion and Consideration of Related Work**

The discussion is generally well-balanced and considers the implications of the findings within the broader field of groundwater hydrology. However, the study could benefit from a more comprehensive comparison with existing research on groundwater responses to droughts in other regions. Citing and discussing more recent studies that have used similar data-driven approaches would strengthen the context of the results.

Response: Thank you for this comment, we will add more references to discuss our work within existing data-driven studies on groundwater droughts. More specifically, we will add

References:

Brakkee, E., van Huijgevoort, M. H. J., and Bartholomeus, R. P.: Improved understanding of regional groundwater drought development through time series modelling: the 2018–2019 drought in the Netherlands, Hydrol. Earth Syst. Sci., 26, 551–569, https://doi.org/10.5194/hess-26-551-2022, 2022.

Chávez García Silva, R., Reinecke, R., Copty, N.K. et al. Multi-decadal groundwater observations reveal surprisingly stable levels in southwestern Europe. Commun Earth Environ 5, 387 (2024). https://doi.org/10.1038/s43247-024-01554-w

Schreiner-McGraw, A. P. and Ajami, H.: Delayed response of groundwater to multi-year meteorological droughts in the absence of anthropogenic management, Journal of Hydrology, 603, 126917, https://doi.org/10.1016/j.jhydrol.2021.126917, 2021

**R2.C2.3  2.3. Appropriateness of References**

The references used in the study are relevant and appropriate, but a few additional sources on groundwater modeling and long-term hydrological trends could be included to further support the study's conclusions. Additionally, ensuring that all citations are up to date would improve the scientific quality.

Response: We plan to add additional recent references regarding long-term trends in groundwater, e.g. Chavez et al. 2024 and Jasechko et al. 2024, also in response to your previous comment. We will also add modelling studies investigating changes in groundwater recharge in Europe and Germany to the discussion, for example

Hellwig, J., Stoelzle, M., and Stahl, K.: Groundwater and baseflow drought responses to synthetic recharge stress tests, Hydrol. Earth Syst. Sci., 25, 1053–1068, https://doi.org/10.5194/hess-25-1053-2021, 2021.

Kumar, R., Samaniego, L., Thober, S., Rakovec, O., Marx, A., Wanders, N., et al. (2025). Multi-model assessment of groundwater recharge across Europe under warming climate. Earth's Future, 13, e2024EF005020. https://doi.org/10.1029/2024ef005020

Wunsch, A., Liesch, T. & Broda, S. Deep learning shows declining groundwater levels in Germany until 2100 due to climate change. Nat Commun 13, 1221 (2022). https://doi.org/10.1038/s41467-022-28770-2

**R2.C3 3. Strengths of the Study**

The study demonstrates several strengths that make it a valuable contribution to groundwater research:

- Extensive Dataset: The study utilizes data from 6,626 monitoring wells, ensuring a comprehensive representation of groundwater variability across Germany.

- Advanced Analytical Techniques: The use of machine learning clustering provides a robust classification of groundwater responses, enabling a more detailed understanding of regional differences.

- Temporal and Spatial Analysis: The research effectively captures the variability in response times, showing how groundwater systems react to droughts over different timescales.

- Relevance to Climate Change Adaptation: The study offers crucial insights for policymakers and water managers, helping to design more effective drought mitigation strategies.

Response: Thank you very much for recognizing these strengths of our study.

**R2.C4 4. Recommendations for Enhancement**

To maximize the impact and accessibility of this research, the following recommendations are suggested:

Response: Please, see our responses below.

**R2.C4.1** 1. Explicitly link cluster classifications to hydrological processes rather than treating them as purely data-driven groupings.

Response: We will improve the accessibility of the clusters through reducing use of acronyms and referring to their meanings, which we will also add in a new Table highlighting the findings to make them more easily accessible at a glance and in connection to the cluster names and properties. Throughout the text, we will expand several formulations to more clearly address the processes represented by the clusters. However, we would like to note that although we identified characteristics more prevalent in certain clusters that reflect underlying dominant processes, these linkages are non-unique, thus naming clusters after the mechanisms would oversimplify the complexity of interacting processes and the ambiguity in the linkages. For this reason, the clusters are named after the observed patterns. Please, also refer to Comment R1.C1.1 from reviewer 1 and R2.C4.2.

**R2.C4.2** 2. Ensure consistency in terminology and reduce the use of acronyms to make the study more accessible.

Response: We will carefully go through the manuscript and improve the accessibility through reducing the use of acronyms while including their meanings and checking for consistency.

Please also refer to your previous comment R2.C4.1 and R1.C1.

**R2.C4.3** 3. Provide a clear justification for the aggregation of groundwater data to a monthly scale, discussing any potential limitations.

Response: We will extend the paragraph describing the data selection. We apologize for any confusion that the description of the data may have caused. We used monthly data as this is the aggregated data provided by CORRECTIV while higher resolved time series were not available. They made this aggregation from the original data with different temporal resolutions, partly daily. At the same time, we think that this temporal resolution is highly suitable to investigate groundwater processes that are temporarily buffered, in particular for event time scales relevant for droughts and their propagation into the groundwater component, e.g. months to years (e.g., Van Loon, 2015). Please, also refer to our response to R1.C2.

**R2.C4.4** 4. Include comparisons with traditional hydrological models to provide additional validation for the machine learning results.

Response: We will add more references to the discussion setting our results into context of existing studies using either hydrological modelling or data-driven approaches (please, refer to Comment R2.C2.2 and R2.C2.3). One of the challenges to compare the large-sample data-driven results to modelling is the that groundwater models across these large scales are rare and hard to parameterize and base on several assumptions and simplifications of the subsurface. We thus think that the different methods and findings complement across studies, but covering hydrological modelling is beyond the scope of this study.

**R2.C4.5** 5. Clarify the significance of R² values and explain their implications for model reliability.

Response: This is a good point, for which we will add reflections on the model performance to the manuscript. The model performances that were reached are in a similar range of a comparable study by Schuler et al. 2022 predicting autocorrelation lengths in groundwater levels in Ireland. Moreover, we think they are not surprisingly low as the predictors used are simple metrics/proxies which are not able to represent the whole complexity of subsurface processes, however, the tendency they show is clear and consistent across the different RF models. This provides confidence in the reliability of model results. Please also refer to the response to R1.C12.

 *"The similar rankings provide confidence in the robustness of results even if model performances are not high in the regression models with R2<0.5. This range of performance is, however, not surprising given the heterogeneity of subsurface conditions and complexity of processes, which cannot be fully represented by the simple characteristics used as predictors."*

**R2.C4.6** 6.Summarize key findings in tables and figures to facilitate quick interpretation of results for a broader audience.

Response: Thank you for this suggestion, we will add a new Table providing a summary of key results across clusters such as medians and majorities of selected characteristics and the connection between cluster names and characteristics (refer also to comments R1.C11, R2.C4.1, R1.C1.1). In addition, we will modify Figure 4 adding labels to guide the reader:

[Figure]

**R2.C5  6. Conclusion and Final Decision**

Overall, the study by Ebeling et al. (2024) is a highly valuable contribution to groundwater research, offering a detailed analysis of aquifer responses to drought events across Germany. However, refining the interpretation of results, improving accessibility, and providing stronger contextualization in traditional hydrological frameworks would enhance the study's impact.

Final Decision: Accepted subject to minor revisions.

The study is well-conceived and presents significant scientific advancements. However, minor revisions are required to improve clarity, ensure methodological transparency, and strengthen the discussion by incorporating additional references and comparisons with previous work. Addressing these aspects will enhance the study's accessibility and scientific rigor.

Response: Thank you for the overall positive feedback. We will carefully go through the manuscript to improve clarity, transparency, and accessibility as well as to enhance the comparison to other works and context. Please, refer to the responses to the specific comments.